# Data-Juicer Sandbox: A Feedback-Driven Suite for Multimodal Data-Model Co-development

Daoyuan Chen [* 1]  Haibin Wang [* 1]  Yilun Huang [* 1]  Ce Ge [1]  Yaliang Li [1]  Bolin Ding [1]  Jingren Zhou [1]

## Abstract

The emergence of multimodal large models has advanced artificial intelligence, introducing unprecedented levels of performance and functionality. However, optimizing these models remains challenging due to historically isolated paths of model-centric and data-centric developments, leading to suboptimal outcomes and inefficient resource utilization. In response, we present a new sandbox suite tailored for integrated data-model co-development. This sandbox provides a feedback-driven experimental platform, enabling cost-effective iteration and guided refinement of both data and models. Our proposed "Probe-Analyze-Refine" workflow, validated through practical use cases on multimodal tasks such as image-text pre-training with CLIP, image-to-text generation with LLaVA-like models, and text-to-video generation with DiT-based models, yields transferable and notable performance boosts, such as topping the VBench leaderboard. A comprehensive set of over 100 experiments demonstrated the suite's usability and extensibility, while also uncovering insights into the interplay between data quality, diversity, model behavior, and computational costs. All codes, datasets, and models are open-sourced to foster future research and applications that would otherwise be infeasible due to the lack of a dedicated co-development infrastructure.

## 1. Introduction

The advent of multimodal large models has revolutionized artificial intelligence, pushing the boundaries of functionality and creativity across various domains (OpenAI, 2024a; Wang et al., 2024a). Recognizing the key role of data in shaping model performance, there are fast-growing efforts to curate datasets of larger scales and higher quality (Jakubik et al., 2024; Gadre et al., 2023; Xu et al., 2024a).

However, the development trajectories of these models and datasets have historically diverged, guided more by intuition than by systematic co-development methodologies. Recent advances in enhancing multimodal large models tend to be either model-centric or data-centric, rarely bridging these two aspects closely. For example, model-centric methods focus on algorithmic enhancements and architectural innovations under fixed data priors, while data-centric strategies usually focus on processing datasets independently of specific model training contexts (Qin et al., 2025). Both approaches usually suffer from a lack of systematic principles and cooperative synergy, relying heavily on heuristic exploration and single-perspective expertise. This fragmented landscape presents a notable barrier to achieving optimal performance, as the interplay between data characteristics and model capabilities remains largely under-exploited.

Moreover, putting multimodal large models into practice is further complicated by infrastructure limitations, increasing computational costs, and the faster pace of the project development (Xu et al., 2024c). In the age of large models with rapidly growing model parameters and dataset sizes, the processes of data processing and model training become increasingly resource-intensive, demanding substantial time and computations. Due to the absence of cost-effective platforms that simplify and speed up data-model co-development, researchers and developers often face the dilemma of prioritizing result-driven development at the expense of thorough, insight-led exploration. This deficiency hinders the iterative refinement for both domains, leading to sub-optimal outcomes as improvements in one domain are hard to inform, apply and enhance each other directly.

To fill this gap, we introduce the Data-Juicer Sandbox, a feedback-driven suite for facilitating the co-development of multimodal data and models. Building upon an open-source data processing system tailored for multimodal large models, Data-Juicer (Chen et al., 2024a;b), our sandbox suite integrates a wealth of off-the-shelf components optimized for usability and compatibility with open-source model-centric infrastructures. Collectively, it offers customizable orches-

---

[*]Equal contribution [1]Alibaba Group. Correspondence to: Yaliang Li <yaliang.li@alibaba-inc.com>.

*Proceedings of the $42^{nd}$ International Conference on Machine Learning*, Vancouver, Canada. PMLR 267, 2025. Copyright 2025 by the author(s).

tration from different levels including end-to-end workflows, specific development behaviors, and underlying data-model development capabilities. Within the sandbox laboratory, users are empowered to optimize data and models based on their fruitful feedback in a cost-controlled environment. This accelerates insight discovery and informed decision-making, paving the way for transferable, resource-efficient data-model optimization.

To exemplify the efficacy of the sandbox, we propose a "Probe-Analyze-Refine" workflow, crafted to explore the interplay between data processing operators (OPs), model performance feedback, and the transferability of these enhancements. We apply this workflow to five cutting-edge models: Mini-Gemini (Li et al., 2024b) and InternVL-2.0 (Chen et al., 2024f), two LLaVA-inspired models for image-to-text generation, EasyAnimate (Xu et al., 2024b) and T2V-Turbo (Li et al., 2024a), two Diffusion Transformer based models for text-to-video generation, and a CLIP model (Gadre et al., 2023) for image-text pre-training. Through over 100 experimental runs and comparative analysis across different tasks, scales and models, we show notable evidence of the Sandbox's usability, including topping the VBench (Huang et al., 2024) leaderboard with superior performance over competitors such as Gen-3 (RunwayML, 2024) and VEnhancer (He et al., 2024a). The improvement is underpinned by a series of insights linking more than 40 data processing OPs and 70 benchmark metrics, with fine-grained analysis on the balance between data quality, diversity, compute cost, and model performance, such as scaling behaviors derived from 1B to 26B mode scales.

Our contributions can be summarized as follows:

- To the best of our knowledge, this is the first open-source [1] sandbox suite tailored for co-development between multimodal data and models, rendering experimental exploration in this field more insightful and systematic.

- We present a new effect-proven workflow for data-model co-development and substantiate its impact through empirical evidence among image understanding, video generation, and image-text pretraining tasks.

- We conduct extensive experiments on benchmarking the effects of dozens of data processing operators and model metrics, providing practical guidance toward further advancements in multimodal large models.

## 2. Related Works

**Model-Centric Progress in Multimodal Large Models.** Multimodal large models have gained prominence for their remarkable capabilities (OpenAI, 2024a;b). Advances in

training algorithms (Caffagni et al., 2024; Li et al., 2024a) and model architectures (He et al., 2024a; Yin et al., 2024) have fueled this interest. Transformer scaling remains a prevalent approach (Xu et al., 2023), though high computational demands and optimization challenges often restrict insights to specific datasets and create a gap in understanding how implicit data biases affect model performance.

**Trends in Data-Centric Development.** Recently, a shift towards data-centric development has emerged (Jakubik et al., 2024), emphasizing data handling as key to the efficacy of large models such as CLIP (Gadre et al., 2023). Despite the increasing recognition of data processing, the heterogeneous nature of multimodal data leads to predominantly heuristic approaches (Long et al., 2024), underscoring the pressing need for more systematic methodologies for data-model co-development.

**Open-Source Infrastructure.** The ecosystem for multimodal model development has expanded with fruitful open-sourced frameworks (Wolf et al., 2020; Liu et al., 2024b). However, contributions to multimodal data infrastructure often consist of raw datasets and preprocessing scripts, lacking standardized practices. Most existing data processing frameworks are tailored for single-modal data (Weber et al., 2024), highlighting the early stage of multimodal data development. To address these gaps, our work presents a new intermediary layer that connects advanced model infrastructures with the Data-Juicer system, facilitating better co-development between models and data.

We present more detailed comparisons of more related works in Appendix A.

## 3. The Proposed Sandbox Laboratory

### 3.1. Motivation and Overview

**Why do we need data-model co-development?** In the era of large models, the development of both data and models necessitates collaboration involving numerous algorithm researchers and system engineers. Training data for large models is often highly heterogeneous in terms of quality, context, type, and timeliness. The processing and mixing of these datasets, known as *data recipes*, are complex and varied (Ge et al., 2024). The scale of data amplifies the stakes for refinement attempts on both data and models, imposing substantial computational and time burdens. Traditional data-centric or model-centric strategies fall short by optimizing in isolation, leading to *diminished overall efficiency* and *resource misallocation* (Qin et al., 2025). When either component requires adjustment, the dual optimization challenge inflates costs, as one part may have already reached near-optimal status.

**Why do we need a sandbox laboratory?** Given the high

---

[1] modelscope.github.io/data-juicer/en/main/docs/Sandbox.html

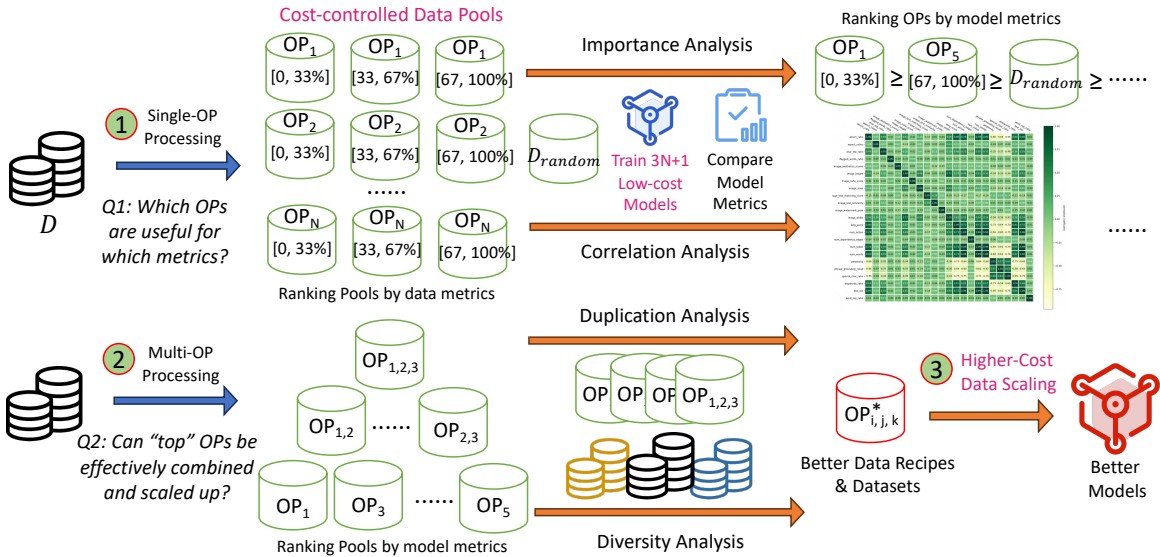

Figure 1: A probe-analyze-refine workflow within Data-Juicer sandbox for systematic data-model co-development.

cost associated with iterative development, existing methods often resort to heuristic approaches for improving data or models. For example, scaling up "cleaned" datasets can be problematic because determining what constitutes a "clean" dataset and measuring its quality qualitatively remains a challenge. Further, the impact of iterations on data and models is difficult to attribute due to numerous influencing factors and considerable engineering effort required. A unified sandbox environment enables experimentation with guided optimization. It permits users to iterate both data and models based on comparative feedback from small-scale data processing and model training trials, thus helping to mine insights that can be transferred to full-scale production environments and increasing the return on investment in computation and time costs.

**Layered orchestration of Data-Juicer sandbox laboratory.** Due to the current absence of ready-made open-source middleware infrastructure, we first propose the necessary codebase and suite, designed with decoupled components. They serve for data analysis, processing, recipe optimization, model training, and evaluation, all managed through configuration files and divided into three levels: (1) In the top level, the specific *workflows* are formed as *ordered job list* for co-development, organized into four phases—probing data/models, refining data recipes, executing data/model operations, and evaluation. One can flexibly adjust task sequences in these phases, reuse built-in workflows, or create their own easily. (2) In the middle level, the common *behaviors* are formed as *hook functions*; and (3) in the bottom layer, the necessary *capabilities* are formed as *factory classes*. The lower two layers work together to facilitate **actionable** development capabilities and pro-

vide **measurable** feedback. Specifically, we simplify the interfaces offered by Data-Juicer, which provides over 100 OPs and tools for multimodal data analysis, filtering, and synthesis, with industry-level optimization for large-scale processing (Chen et al., 2024b). Built upon it, we integrate many SOTA open-source libraries for model training and evaluation such as Mini-Gemini, EasyAnimate, ModelScope, VBench, MMBench (Liu et al., 2024b), TextVQA (Singh et al., 2019) and MME (Fu et al., 2023).

Note that the underlying libraries are still evolving, and the sandbox is continuously maintained to incorporate more community efforts in other libraries, optimized for usability and reduced cognitive load. Dedicated scripts combine the classes and hooks into specific workflows, with code and parameters adjusted to ensure a cost-controlled environment. This suite serves as a groundwork for the deriving following experimental insights. More details from the infrastructure perspective are in Appendix B.

### 3.2. A Probe-Analyze-Refine Workflow

To demonstrate the usage of the sandbox, we introduce a structured workflow illustrated in Fig. 1. It is designed to answer several key questions for informed decision-making and cost-effective data-model co-development:

(1) Given the variability in data, models, and application scenarios, we initially seek to find out *which data processing OPs contribute most effectively to enhancing model performance* (Sec. 3.2.1). This involves creating equal-size data pools, each processed uniquely by a single OP and subsequently sorted by interested data feedback signals. Reference models are trained on these data pools at low

and controlled costs. This step enables us to analyze and understand OP impact and correlations with models.

(2) Guided by insights derived from the most impactful OPs that ranked highest by model feedback signals, we proceed to study *whether these OPs can be effectively combined into data recipes and scaled up* (Sec. 3.2.2). We establish a hierarchical data pyramid, wherein data pools are categorized across different tiers based on OP ranks. This step also examines the viability of OP combinations when scaled with increased data volumes.

(3) Built upon found OP combinations, we delve into a dual analysis focusing on data *duplication* and *diversity* (Sec. 3.2.3). We assess whether the model training would benefit *more compute* from the repeated use of high-quality data pools or from the inclusion of lower-quality data to expand the overall data pool. As a result, we obtain data recipes optimized from numerous small-scale comparative experiments, which can then be applied to larger-scale scenarios with reduced amortized cost (Sec. 3.2.4).

### 3.2.1. SINGLE-OPERATOR DATA POOLS

Starting with an initial dataset $\mathcal{D}$, we define a single-OP data pool $\mathcal{P}_i$ as the dataset processed exclusively by the $i$-th OP ($\mathcal{OP}_i$) available in Data-Juicer as $\mathcal{P}_i = \mathcal{DJ}[\mathcal{OP}_i(\rho_i)](\mathcal{D})$, where $\mathcal{DJ}$ denotes the data-processing function implemented by Data-Juicer, and $\rho_i$ is the hyper-parameters governing the operation of $\mathcal{OP}_i$. For example, filter OPs of Data-Juicer compute specific statistics and then apply threshold criteria to select data samples. Within this workflow, $\mathcal{D}$ is processed by $N$ interested filter OPs, the resulted pools $\{\mathcal{P}_i\}$ are sorted by different data statistics and segmented into three equal-sized pools $\mathcal{P}_{i,\text{low}}$, $\mathcal{P}_{i,\text{middle}}$ and $\mathcal{P}_{i,\text{high}}$, representing data with low, middle and high stats, respectively. Besides, $\mathcal{D}$ is randomly sampled to serve as a control group $\mathcal{D}_{rand}$ such that all the $3N + 1$ data pools have the same data size. This design fosters discriminative insights across varying degrees of data processing intensity.

Subsequently, models are trained independently on $\{\mathcal{P}_i\}$ and $\mathcal{D}_{rand}$ with consistent hyper-parameters, data sample size and compute resources, and finally evaluated across interested performance metrics. This linking of feedback signals between data stats and model metrics enables insight mining and top OP identification.

### 3.2.2. MULTI-OPERATORS DATA POOLS

We then apply multiple OPs sequentially (denoted as $\mathcal{S}$) to examine whether these OPs complement or counteract each other's effects. To achieve this, we extend the previous construction of $\{\mathcal{P}_i\}$ to multi-OPs case: $\mathcal{P}_{\mathcal{S}} = \big(\mathcal{DJ}[\mathcal{OP}_i(\rho_i)] \circ \mathcal{DJ}[\mathcal{OP}_j(\rho_j)] \circ \cdots \circ \mathcal{DJ}[\mathcal{OP}_k(\rho_k)]\big)(\mathcal{D})$, where $i, j, ..., k \in \mathcal{S}$, indicating $n_s$ candidate OPs used for

sequential combination.

The number of possible $\mathcal{S}$ grows exponentially with $n_s$, necessitating non-trivial selections of combinations to explore. Building on insights from single-OP experiments, where the "Top" OPs are represented by their ranks, we propose two practical strategies: (1) combining "Top" OPs based on their progressively diminishing impacts on model performance, and (2) clustering OPs based on their Pearson correlation coefficients and then combining the "Top" OPs within each category. Similar to the single-OP scenario, we then consistently train reference models on $\{\mathcal{P}_{\mathcal{S}}\}$ and $\mathcal{D}_{rand}$ to gain comparative outcomes. More implementation details and empirical results on these two strategies can be found in Appendix D.1 and Sec. 4.3 respectively.

### 3.2.3. PYRAMID-SHAPED DATA POOLS

Adopting a larger $n_s$ may lead to enhanced data quality while less available data volume. This phenomenon prompts an investigation: should we prioritize reusing high-quality data or incorporate lower-quality yet more abundant data to escalate training compute scales?

To embed this inherent trade-off between data quality and diversity, we devise a hierarchical pyramid architecture, where $n_s$ "top" OPs identified in single-OP experiments combine into $2^{n_s}-1$ data pools. For example, the three-OP combination $\mathcal{OP}_{1,2,3}$ resides at the highest level of the hierarchy but has the smallest data volume after filtering. The two-OP combinations, such as $\mathcal{OP}_{1,2}$, are placed at a lower level and may have volumes several times larger than $\mathcal{OP}_{1,2,3}$. Progressing downward through the pyramid, data pools exhibit a descending average OP ranking (which potentially acts as feedback of reduced data quality) alongside an increase in data volume.

To strike the balance between data quality, diversity and compute scale, we consider two settings built upon this data pyramid: (1) repetitive training with data repetition from the top-layer data pools, and (2) non-repetitive training via progressively adding lower-layer data pools and applying deduplication. Specifically, we explore variable repetition rates for the first setting and make the number of trained data samples of the second setting consistently match the former. It allows us to qualitatively assess the efficacy of data reuse compared to the inclusion of suboptimal data, within the same compute costs at varying scales.

### 3.2.4. DISCUSSION ON SCALING AND COST

All data pools are uniformly sampled and consistently derived from $\mathcal{D}$, enabling insights gained from small-pool experiments to be extrapolated to larger-scale scenarios, and making the method capable of overall cost reduction.

To clarify, denote the model training time using the full

dataset $\mathcal{D}$ as $T_{full}$. Let $M$ represent the iterations required to achieve desirable performance without our sandbox. Let $T_{pool}$ represent the training time with a small pool, which is reduced by a ratio of $r$ compared to $T_{full}$. If the total number of planned small-pool experiments is $m$, then the total time without and with our workflow is $M \times T_{full}$ and $(T_{full} + m \times T_{pool}) \approx (1 + mr) \times T_{full}$, respectively. Our core idea is to construct $m$ lightweight experiments and utilize fine-grained data-model feedback to extract insights for scaling and reducing overall costs, making it preferable to disjoint model- or data-centric development that involves $M$ costly (and usually heuristic) experiments.

Achieving $(1 + mr) \leq M$ is feasible. Typically, $M > 2$ accounts for multiple iterations over different model and dataset versions. The $m$ positively corresponds to the interested OPs, often numbering in the dozens. The $r$ can be much smaller than 0.01, since there is no need for small-pool experiments to "converge" as long as they reveal informative changes—positive or negative—in model performance after controlled data interventions with $\{\mathcal{P}_i, \mathcal{P}_S\}$ versus baselines using $\mathcal{D}_{rand}$. Early stopping of unpromising experiment trials can also help expedite this process.

Furthermore, $m$ reflects the balance between the intensity of data interventions and the informativeness of data-model feedback, offering flexibility to adjust the experimental plan based on available resources. As for the factor $r$, we have

$$\mathbb{P}[\Delta_{pool} - \mathbb{E}[\Delta_{full}] \geq \epsilon] \leq e^{-2\epsilon^2/(b-a)^2},$$

where $\Delta_{pool} \in [a, b]$ and $\Delta_{full}$ denote the model performance change with our small and full data pools respectively. This indicates that the error $\epsilon$ introduced by our workflow decreases exponentially as $r$ increases. More detailed analysis is provided in Appendix C.

## 4. Practical Applications and Main Results

### 4.1. Use-Cases and Experiment Overview

To demonstrate the proposed suite's usability and effectiveness, aiding understanding and boosting confidence in its applicability, in this section, we conduct extensive experiments in various practical use cases. They vary in models and data OPs (Sec. 4.2 and Sec. 4.3), data samples and compute cost (Sec. 4.4), leading to insights and recipes useful for larger-scale scenarios (Sec. 4.5).

- For image-to-text (**I2T**) generation, we will show that the optimal recipe derived from small pools achieves superior model performance and higher data efficiency on the MGM model when applied to larger datasets.

- For general text-to-video (**T2V**) generation, we will show that the insights derived from small pools and the EasyAn-

imate model can lead to VBench Top-1 performance with another architecture-different model.

- For image-text pre-training (**ITP**), we will show that the optimal recipe identified with a CLIP model having fewer FLOPS maintains consistent performance advantages when model FLOPS and compute resources increase, similar to scaling law behaviors.

- For image-captioning (**IC**) oriented fine-tuning, we will show that the optimal recipe identified with InternVL-2.0 model (Chen et al., 2024f), leads to consistent and steady performance advantages as the model scale increases from 1B to 26B.

- To demonstrate the flexibility, extensibility, and potential of the proposed sandbox, we further conduct experiments on "iterative workflow", "OPs beyond filters", and "model development of automated prompt adjustments". More details will be introduced later in Sec. 4.6.

These examples instantiate different behavior hooks and capability classes introduced in Sec. 3.1, following the workflow proposed in Sec. 3.2. A more structural summarization of these use cases is provided in Table 5 in Appendix E.1.

In the next subsections, we delve into the primary insights, with complete experimental results provided in Appendix E. The feedback utilized in these experiments comes from over 70 widely-used model benchmark measurements and over 40 Data-Juicer's data filtering OPs for text, image, video, and cross-modalities. For brevity and comprehensive investigation, on the main page, we report the performance changes relative to baselines on $\mathcal{D}_{rand}$, averaging from all evaluated benchmark scores. For a fair comparison, note that all experiments are maintained to ensure that the reference models trained on $\mathcal{P}_i$ and $\mathcal{P}_S$ have the same compute costs to baselines trained on $\mathcal{D}_{rand}$. Detailed task-specific settings can be found in Appendix D, including the benchmark measurements (in Appendix D.2), the functionalities of studied OPs (in Appendix D.3), and the specific sources and sizes of $(\mathcal{D}, \mathcal{D}_{rand}, \mathcal{P}_i, \mathcal{P}_S)$, in Appendix D.4 to D.6.

### 4.2. Ranking Single-Operator Data Pools

Table 1 summarizes the results of reference models trained on top-performing data pools. The full tables of all studied OPs can be found in Appendix E.2.

> **Observation 1    (Data vs. Model)**
>
> Multimodal models' efficacy is closely tied to the fidelity of their output modalities, which can be explicitly reflected in the filtering of input training data.

For the text-to-**video** task, all top-3 OPs are video-only OPs. For image-to-**text** and **image-text** pre-training tasks, two of

Table 1: The average performance changes relative to baseline models on $\mathcal{D}_{rand}$ for top-3 performing OPs, with models trained on different single-OP data pools split by sorting OP-generated statistics: $\mathcal{P}_{i,low}, \mathcal{P}_{i,mid}, \mathcal{P}_{i,low}$. The training data pool for baseline is randomly sampled, and all compared pools are with equal data volume. Full ranking table can be found in Appendix E.2.

| Task | OP Statistics | Avg. Perf. Changes (%) | | |
|------|---------------|:---:|:---:|:---:|
| | | $\mathcal{P}_{i,low}$ | $\mathcal{P}_{i,mid}$ | $\mathcal{P}_{i,high}$ |
| | *Image NSFW Filter* | 7.13 | 18.44 | **66.38** |
| I2T | *Text Action Number* | **59.90** | 0.29 | -2.05 |
| | *Language Score* | **49.90** | 0.85 | -1.43 |
| | *Video Aesthetics Score* | -0.98 | 0.13 | **0.96** |
| T2V | *Video NSFW Score* | **0.82** | -0.05 | -0.57 |
| | *Frames-Text Similarity* | -1.45 | 0.23 | **0.79** |
| | *CLIP Image-Text Similarity* | -32.57 | -6.39 | **39.53** |
| ITP | *BLIP Image-Text Similarity* | -24.28 | 1.82 | **25.39** |
| | *Image NSFW Score* | **12.18** | 1.28 | -18.38 |
| | *Text Length* | **0.76** | -3.13 | -11.36 |
| IC | *Image Watermark Score* | -0.64 | -0.13 | **0.48** |
| | *Character Repetition Ratio* | **0.45** | -0.46 | -0.63 |

their top-3 OPs are text-only, and image-text OPs respectively. This trend of influential OPs holds beyond top-3 ranks (detailed in Appendix E.2), suggesting that more attention and resources should be allocated to data processing related to the output modalities of the studied models.

---

**Observation 2    (Diversity vs. Quality)**

In contrast, data diversity is more crucial for image-to-text models, while data quality is key for text-to-video and image-text pre-training models.

---

We find that some top OPs share similar functionalities, while their best-performing pools exist in different statistical ranges. For example, $\mathcal{P}_{i,high}$ in image-to-text, and $\mathcal{P}_{i,low}$ in text-to-video and image-text pre-training. A deeper analysis of OP ranks in terms of *NSFW scores* and *language scores* reveals that the studied image-to-text model prefer more diverse data compared to the text-to-video and image-text pre-training models. Intuitively, high-scoring images or videos in *NSFW* (Not Safe For Work) content are generally rare and occupy the long tail of the data distribution. Consequently, pools with high *NSFW scores* tend to be more diverse. Similarly, texts with low *language scores* indicate that the language of the text is more difficult to identify and usually less common, corresponding to more diverse pools. Appendix E.3 presents more quantitative evidence in terms of word entropy on these data pools' captions.

---

**Observation 3    (Spatiotemporal Dynamics)**

Dynamic information in data is harder to learn for image-to-text and image-text pre-training models than for text-to-video models.

---

Due to the static nature of images, image-to-text and image-text pre-training models often struggle with dynamic content, requiring in-depth semantic understanding. This is evident by their better performance on data pools with fewer *text action numbers*. In contrast, text-to-video models perform differently, showing opposed trends preferring higher *text action numbers* and *video motion scores*.

---

**Observation 4    (Modality Alignment)**

High data alignment between modalities is crucial for the multimodal tasks image-to-text, text-to-video, and image-text pre-training.

---

Image-to-text and text-to-video models prefer modality-aligned data, as indicated by their good performance with pools featuring high *image-text similarity*, *phrase grounding recall*, and *frames-text similarity*. This preference is most pronounced in image-text pre-training models, whose top-2 OPs are tied to image-text alignment.

### 4.3. Shaping Data Recipes of Top OP Combinations

Based on top-3 OPs in Table 1 and gained insights, we then study the model performance changes trained on multi-OP recipes $\mathcal{P}_S$ over baselines on $\mathcal{D}_{rand}$.

---

**Observation 5    (Effect of Sequential Combination)**

The optimal data recipe isn't always achieved by combining the best individual OPs, nor does adding more high-performing OPs guarantee better outcomes.

---

As shown in Fig. 2, combining higher-performing OPs does not always yield better results. For instance, in the image-to-text experiment depicted in Fig. 2(a), the data pool with a high *image NSFW score*, despite performing best in single-OP experiments, generally diminishes performance when combined with others. Similarly, in Fig. 2(b), while pairwise combinations of OPs show positive gains, integrating the top-1 OP into the combination of the other two OPs reduces the relative improvement from 2.48% to 1.88%. As for the image-text pre-training tasks (in Fig. 2(c)), using the `image_text_similarity_filter` OP alone outperforms other combinations. This observation may be attributed to the filtering process relying on a high-quality CLIP model to train another CLIP model, implicitly and effectively acting as a form of model distillation.

Besides, we explore categorizing OPs based on analysis of their stats correlations (Appendix E.4) and select the



Figure 2: The model performance changes (%) from recipes combined with the top-3 OPs listed in Table 1.

optimal OP from each category to form recipes. However, detailed results of Observation 8 in Appendix E.5 show that it also yields non-positive outcomes. These two observations challenge the common assumption in many existing SOTA works that stacking multiple intuitively useful data-cleaning actions can enhance overall performance.

> **Observation 6** (Effect of Seed OPs)
>
> Single OP performance positively correlates with the performance of multi-OP recipes containing it. Starting with high-performing OPs is a good initial step to optimize higher-order data recipes.

Although the Top-3 OP recipes exhibit suboptimal performance, we observe positive gains when combining some pairs of OPs in them can outperform both single top-1 and top-3 combinations, such as in the case of the image-to-text task when combining `TextActionFilter` and `LanguageIDScoreFilter`.

## 4.4. Scaling Data Samples, Computes & Model Size

Recall that in Sec. 3.2.3, we proposed constructing pyramid-shaped data pools by combining different "top" OPs and explored how to scale up the data samples and computes by either reusing high-quality top pools or by progressively adding lower-level pools to enhance diversity with deduplication. Here we select top-1 and top-2 recipes identified in Fig. 2 as candidate data pools within the pyramid structure. The results are summarized in Fig. 3, where the expansion rate indicates the compute scale in terms of the number of training data samples.

> **Observation 7** (Effect of Compute Scaling)
>
> Scaling up compute resources with high-quality data benefits both the image-to-text, text-to-video, image-text pre-training, and image-captioning tasks. Among these, the image-text pre-training and image-captioning tasks demonstrate clear scaling law behaviors in exponential form across different dataset and model sizes.

Generally, the models trained with data derived from top recipes (red lines) achieve superior performance over base-

lines, including those incorporating data from suboptimal recipes. Specifically, for the **image-to-text** (see Fig. 3(a)) model, duplicating top-quality data by a factor of 6 results in higher performance than both using 8x compute with suboptimal data (blue line) and using 8x compute with the original full-size data (green line). For the **text-to-video** model (see Fig. 3(b)), the benefits of using top-quality data outweigh detrimental effects until 10x data repetitions, showing near-linear and significantly higher performance improvements compared to the baseline (blue line).

For the **image-text pre-training** task, Fig. 3(c) illustrates improvements as both model size and compute scale increase. Notably, the linear growth in relative improvement with the exponential increase in computation aligns with known scaling laws (Cherti et al., 2023), suggesting that expanding model size and compute resources, alongside processing data via our high-quality recipes, yields consistent scaling of performance gains.

Moreover, we conduct experiments using InternVL2.0 across various scales (1B, 2B, 4B, 26B parameters), with the results summarized in Fig. 3(d) (x-axis displayed on a log scale). We initially experimented with the smallest model (1B parameters) on 23 single-OP and 6 multi-OP data pools to identify the optimal data recipe. The top-3 combination recipes derived from this initial phase were subsequently applied to all selected model scales. Our findings indicate that all three recipes consistently maintain significant performance advantages as the model scale increases from 1B to 26B parameters, demonstrating clear scaling law behaviors.

Table 2: Average performance of MGM-2B with different pretraining datasets on 4 benchmarks. The "*" indicates our reproduced version that is comparable with the official version. Detailed results are in Appendix E.6, Table 10.

| MGM-2B | Num. of Training Instances | Avg. Perf. Changes (%) |
|---|---|---|
| Baseline* | 1226k | - |
| Ours | **159k (x4)** | **+2.12** |

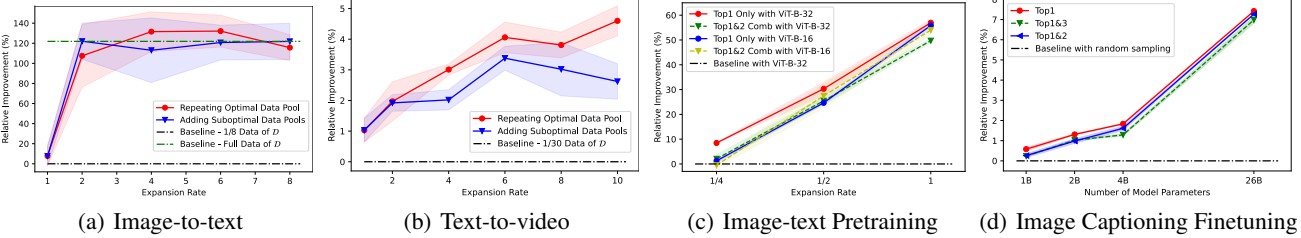

Figure 3: Relative improvement over baselines when varying computation scales with and without data duplicates. In the latter sub-figures, "Top1" and "Top2" refer to applying the Top-1 and Top2 recipes identified in Sec 4.3 respectively.

### 4.5. Transferring Recipes to Larger-Scale Heterogeneous Scenarios

The previous **image-text pre-training** and **image-caption fine-training** experiments, conducted on CLIP and InternVL models of varying sizes (Fig. 3(c) and Fig. 3(d)), already shown that top-performing recipes identified at small scales can be effectively utilized when compute resources are scaled up. In this section, we further apply the gained insights and top data recipes to larger-scale heterogeneous models (i.e., across different model architectures) and heterogeneous datasets (i.e., from diverse data sources) for the other two tasks (image-to-text and text-to-video).

**The case for image-to-text task.** We follow Observations 6 and 7, and utilize the top data pool from Fig. 2(a) with a 4x repetition, resulting in a pre-training dataset of 637k samples, which constitutes only half of its original pre-training dataset size. Then we train the MGM-2B model using this new pre-training dataset and its original full fine-tuning dataset for comparison.

Table 2 summarizes the results. Since the official MGM-2B model has not been evaluated on MMBench-CN, we trained a reproduced version using the official training scripts, and its performance is closely aligned with the benchmarks reported for the official model. We can see that compared to the baseline trained on the full dataset, our model achieves superior performance while trained on only 1/10 of the distinct instances and 1/2 of its original total instances (i.e., using half of the baseline's compute FLOPs). Notably, they differ only in the processing of the pre-training data, with the model and training configurations being consistent, underscoring the effectiveness of our insights.

**The case for text-to-video task.** In line with the best recipe from Fig. 2 and Observation 6, we scale up the data pool on the full-size candidate video datasets introduced in Sec. D.5, with low *video NSFW score* and high *frame-text similarities*.Additionally, following Observation 7, we conduct six model training passes through this dataset. From a data development view, this transition allows us to probe the scalability of our methodologies, advancing from the small pool

with 40k data samples used in Sec. 3.2.2 to a 5.7× larger pool with 228k data samples. From a model development view, we undertake a challenge to assess the transferability of our findings across different model architectures, by replacing the training model from the previous EasyAnimate with another SOTA model, T2V-Turbo, which is improved from VideoCrafter-2.0 (VC2).

Table 3 showcases our notable performance on the VBench leaderboard, reporting average scores of 16 metrics from both quality and semantic dimensions. We first enhance T2V-Turbo (the last row) with data-enhanced distillation training on 147k instances (the second row), and then self-distill it with other 228k instances (the first row). Compared to the base model in the last row, our method yields notable uplifts of 1.53% and 2.59% on the *Board* and *Uniform Average* scores respectively. This verifies the effectiveness of our sandbox and insights again, which leverages `VideoFramesTextSimilarityFilter` OP to enhance the video-prompt alignment, and the `VideoNSFWFilter` OP to ensure the high quality of the generated video.

Note that these results highlight the data-efficiency of our method to reduce overall cost with dedicated data-model co-development. Taking the third-place model, VEhancer, as an example, it mainly focuses on architectural refinements building upon the foundation of VC2. While our model also originated from VC2, it gains a higher performance boost but with at least 22x less compute costs than VEhancer. Detailed calculation of FLOPs is presented in Appendix E.9. Besides, in Appendix E.7, we further conduct an ablation study to verify the effectiveness of our data-model co-improvement over the base model T2V-Turbo, including aspects from data (high-quality dataset with our derived recipes), model (LoRA initialization and self-distillation), and data-model co-design (real-data loss and self-evolution). These results demonstrate the potential cost-effectiveness of the proposed sandbox when utilizing fine-grained feedback from both the data and the model simultaneously.

Table 3: Models on the VBench leaderboard as of Sep 23, 2024. "Board Avg." denotes the weighted average scores across 16 metrics defined by VBench and "Uniform Avg." denotes the arithmetic average. "$-^{\dagger}$" and "$-^{\ddagger}$" indicates proprietary model, and acting as our base teacher model respectively. $\alpha$ indicates a constant for specific gradient update implementation. Detailed ablation study, full ranking results, and FLOPs estimation are in Appendix E.7, E.8 and E.9 respectively.

| Models (Ranked by leaderboard) | Board Avg. (%) | Uniform Avg. (%) | Dataset Size | Training Samples | Training Cost (EFLOPs) |
|---|---|---|---|---|---|
| 1. **Data-Juicer (DJ, 228k)** | **82.53** | **81.26** | 228k | 640k | $7.68\alpha$ |
| **Data-Juicer (T2V, 147k)** | 82.10 | 80.54 | 147k | 640k | $7.68\alpha$ |
| 2. Gen-3 (RunwayML, 2024) | 82.32 | 79.64 | $-^{\dagger}$ | $-^{\dagger}$ | $-^{\dagger}$ |
| 3. VEnhancer (VC2) (He et al., 2024a) | 81.97 | 80.00 | 350k | $\geq 350\text{k}$ | $\geq 167.3\alpha$ |
| 4. Kling (2024-07) (Kuaishou, 2024) | 81.85 | 79.54 | $-^{\dagger}$ | $-^{\dagger}$ | $-^{\dagger}$ |
| 6. CogVideoX-5B (Yang et al., 2024) | 81.61 | 79.90 | 35M | 35M | $\geq 11841\alpha$ |
| 8. T2V-Turbo (VC2) (Li et al., 2024a) | 81.01 | 78.67 | $-^{\ddagger}$ | $-^{\ddagger}$ | $-^{\ddagger}$ |

### 4.6. Additional Experiment Results and Details

**Further Analysis and Ablation Studies.** Appendix E.3 and E.4 detail the data diversity and correlation analyses on our multiple-OP experiments, respectively. They showcase the analysis ability provided by the suite with word entropy, word cloud visualization, and visualizations of Pearson correlation coefficients among OP statistics and benchmark performance. Appendix E.5 evaluates the performance of models trained using recipes derived from the correlation analysis. Appendix E.7 provides a detailed ablation study of the Data-Juicer T2V models, focusing on data-model co-improvements beyond the base teacher T2V model. This includes analyses from the perspective of data (high-quality datasets with derived recipes), model (LoRA initialization and self-distillation), and data-model co-design (real-data loss and self-evolution).

**Extensibility of the Proposed Sandbox.** Appendix E.10 verifies the capability of iterative workflows, illustrated by the progression: $\text{ckpt}_0 \xrightarrow{\text{refine}} \text{recipe}_1 \xrightarrow{\text{train}} \text{ckpt}_1 \xrightarrow{\text{refine}} \text{recipe}_2 \xrightarrow{\text{train}} \text{ckpt}_2$. Notably, $\text{ckpt}_2$ demonstrated improved performance despite the evolving recipes. Appendix E.11 demonstrates the potential of operators beyond Filters, showcasing two representative Mappers, the `image_diffusion_mapper` and `image_captioning_mapper`, and their integration with the MGM model. Finally, Appendix E.12 demonstrates how the sandbox's pre-built infrastructure can be leveraged to explore and auto-optimize prompts within the context of data-model co-development.

**Complete Results.** As for the complete results of other previously mentioned experiments. Appendix E.2 presents the complete operator ranking from single-operator experiments. Appendix E.6 provides comprehensive results for the Image-to-Text task. The full V-Bench leaderboard results are listed in Appendix E.8. Finally, Appendix E.9 details the calculation of reported FLOPs.

**Implementation Details.** We provide a summary of the varying scope of our sandbox experiments in Appendix E.1, detailing aspects such as main effectiveness evidence, model types, dataset size, and compute scale. Besides, we present a discussion in Appendix D.1 on the two strategies employed to combine multiple operators into recipes based on their correlations. Appendix D.2 outlines the model evaluation metrics adopted. The functionalities and corresponding statistics of the studied Data-Juicer operators are detailed in Appendix D.3. Furthermore, Appendix D.4 elaborates on the detailed settings for the image-to-text use case, Appendix D.5 for the text-to-video use case, and Appendix D.6 for the image-text pre-training use case.

## 5. Conclusion and Future Works

In this paper, we introduced the Data-Juicer Sandbox, a new open-source suite designed to facilitate the co-development of multimodal data and models. By integrating flexible and customizable components at different levels, the sandbox enables systematic, cost-effective exploration and optimization, bridging often-disjoint domains of data and model development. Through applying the proposed "Probe-Analyze-Refine" workflow in extensive scenarios including image-to-text generation, text-to-video generation and image-text pretrianing, we showcased how our sandbox can yield not only improvements in both dataset and models, but also valuable insights into the complex interplay between data processing and model performance.

To facilitate reuse and expedite innovation, we will continuously extend the sandbox's compatibility to encompass more model-centric infrastructures and develop more off-the-shelf and effect-proven workflows, such as for reinforced fine-tuning (Pan et al., 2025). Besides, we will theoretically investigating the trade-off between sandbox cost and feedback transferability, such as taking the sampling ratio of data pools as a tunable knob.

## Impact Statement

This paper presents work whose goal is to advance the field of Machine Learning. There are many potential societal consequences of our work, none which we feel must be specifically highlighted here.

## Reproducibility Statement

Reproducibility is essential for validating research outcomes. To facilitate this, we have organized detailed descriptions within the appendix of our paper. Key components of our experimental setup, including implementation details such as datasets and training configurations for the image-to-text, text-to-video and image-text pre-training use cases can be found in Appendix D.4, Appendix D.5 and Appendix D.6 respectively. We also provide details into the methodologies for combining multiple operators based on their correlations in Appendix D.1, as well as descriptions of performance metrics (Appendix D.2). Furthermore, Appendix D.3 outlines the functionalities and statistics of the Data-Juicer OPs utilized in our experiments.

All codes, datasets, and models of our work are openly accessible and actively maintained at https://github.com/modelscope/data-juicer/blob/main/docs/Sandbox.md.

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

# Appendix

## Table of Contents

## A. Discussion on More Related Works

**Model-Centric Progress in Multimodal Large Models.** Multimodal large models have captivated researchers with their formidable capabilities (OpenAI, 2024a;b), leading to a surge in model-centric development efforts. These focuses mainly lie in refining training algorithms (Caffagni et al., 2024; Li et al., 2024a; Zhang et al., 2024), advancing model architectures and components (He et al., 2024a; Yin et al., 2024; Jiao et al., 2024), and harnessing the models' potential for various applications (Wang et al., 2024a; Liu et al., 2024a; Zhou et al., 2024a). There is a growing consensus that transformer-based scaling is predominant (Xu et al., 2023). However, the high computational requirements imposed by scaling laws (Xu et al., 2024c) and the optimization challenges inherent to large models (Manduchi et al., 2024) often confine insights to specific datasets or vague data characteristics. This situation leaves a significant gap in comprehending the extent to which models' performance and behavior hinge upon implicit assumptions and inductive biases embedded within the underlying data distributions. In contrast, our work demonstrates a feasible and promising path to fill in this gap by explicitly linking data processing effects with the downstream performance of trained models through numerous contrastive sandbox experiments.

**Trends in Data-Centric Development for Multimodal Large Models.** An emerging trend shifts the focal point from models to data (Jakubik et al., 2024; Bai et al., 2024; Chen et al., 2024c; Jiao et al., 2025), underscored by the notion that large models function akin to data compressors (Delétang et al., 2024). Echoing the principle of "garbage in, garbage out", meticulous data processing is recognized as pivotal. Efforts now isolate data manipulation as a primary experimental variable in multimodal generative modeling (He et al., 2024b). Nonetheless, multimodal data processing involves highly heterogeneous processing workflows, vast quantities, diverse types, and the high cost of training downstream models. This complexity results in predominantly heuristic approaches, such as data filtering and synthesis guided by human intuition (Long et al., 2024; Zhou et al., 2024b).

For example, one well-studied model type is CLIP. Data-Comp (Gadre et al., 2023) introduces a benchmark to filter out high-quality data from 12.8 billion image-text pairs in Common Crawl to train better CLIP models, considering Filter Operators such as CLIP score, image size, and caption length. MetaClip (Xu et al., 2024a) aims to reproduce CLIP's data by introducing a raw data pool and finding a balanced subset based on CLIP's concepts. Unlike these data-centric approaches that isolate the model and training settings, focusing solely on the quality and scale of training datasets, our work emphasizes systematic methodologies for data-model co-development, considering both data and

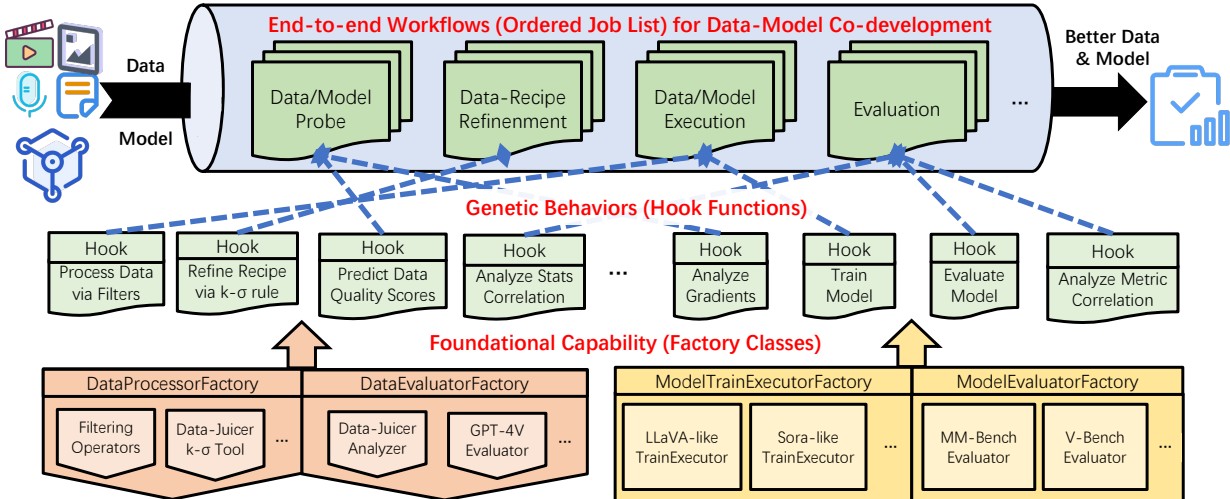

Figure 4: Overview of the Data-Juicer Sandbox Laboratory. The workflow involves four stages, each allowing flexible customization at different levels within the data-model co-development lifecycle. More details can be found in Sec. B

models equally important. Specifically, we incorporate performance signals from sandbox reference models on many downstream tasks, conducting importance and correlation analysis to link data pools and these model metrics. Additionally, we explored more model types beyond CLIP, such as LLaVA-like and DiT-based models for image-to-text and text-to-video tasks, and identified better training datasets for these models using our workflow.

**Open-Source Infrastructure for Multimodal Large Model Development.** The landscape for multimodal model development has advanced significantly, boasting a variety of strong infrastructures and frameworks for training and evaluation. Prominent examples include Transformers (Wolf et al., 2020), Diffusers (von Platen et al., 2022), NeMo (Harper et al.), MMagic (MMagic Contributors, 2023), ESPNet (Peng et al., 2023), and MMBench (Liu et al., 2024b).

However, when it comes to multimodal data infrastructure, the primary contributions have been datasets and dataset-specific preprocessing tools, such as DatasetsHub from HuggingFace (Lhoest et al., 2021). The standardization and efficient utilization of practical expertise and foundational data processing capabilities remain unaddressed. Existing frameworks for data processing predominantly focus on single-modal data (Weber et al., 2024; Bradski, 2000; Hwang et al., 2023), underlining the early stages of development in systematic platforms for multimodal data (Chen et al., 2024a; Du et al., 2023; Chen et al., 2024b).

Recognizing the critical interplay between datasets and models—where comprehensive, high-quality datasets enhance model performance, and advanced models contribute to the generation of even more refined datasets—our work stands

out by introducing an innovative intermediary layer. We seamlessly integrate cutting-edge model-centric multimodal infrastructure with the Data-Juicer system. This integration fosters a streamlined and insightful co-development environment for both models and data, bridging the current gap and setting a standard for future efforts in the multimodal domain.

## B. Sandbox Suite from Infrastructure Perspective

### B.1. Overview Architecture

To support co-development based on feedback from data and model, we design the layered sandbox laboratory as illustrated in Fig. 4. The laboratory incorporates a spectrum of components for activities such as data analysis, filtering, recipe optimization, model training, inference, and evaluation, all orchestratable via configuration files. The architecture is stratified into three tiers: bespoke end-to-end *workflows*, generic development *behaviors*, and foundational data-model development *capabilities*.

- The top tier delineates co-development workflows executed sequentially across four phases: probing data/models, refining data recipes, executing data/model operations, and evaluation. The sequence of tasks within each phase is adjustable through an input configuration file, permitting users to leverage pre-established and effect-proven workflows or customize their own with ease.

- Moreover, users can flexibly introduce or innovate classes at the capabilities level (such as novel models, metrics, or data processing algorithms) and behavior hooks (like mul-

tidimensional data quality assessments and adaptive adjustments based on multiple probe outcomes) interchangeably.

This design allows for streamlined configuration and reuse of established infrastructure. Importantly, it expedites the prototyping of data and model development solutions, integrating actionable and measurable capabilities for swift feedback derivation and informed decision-making. Concrete explorations of the lower tiers are provided in Sec. B.2, while Sec. 3.2 showcases the sandbox's utility in establishing a probe-analyze-refine workflow for data-model co-development.

### B.2. Flexible Capability Factory and Behavior Hooks

**Actionable Perspective.** Within the sandbox, we have created various factory classes and behavior hooks for data processing. These simplify and unify the interfaces offered by the open-source system, Data-Juicer. Users can utilize over 100 OPs and tools for data analysis, filtering, and synthesis. The toolkit automatically speeds up and scales up the processing through system optimization and parallel support. Classes for refining data recipes leverage tools like adjusting percentile distributions to distinguish data subsets or applying k-sigma rules to remove outliers.

Besides, model development classes integrate SOTA open-source libraries, streamlining interfaces and enhancing usability for rapid and user-friendly development experiences. For example, models can be easily trained with diverse functionalities, such as Mini-Gemini for image-to-text, EasyAnimate and T2V-Turbo for text-to-video, and ModelScope for general generative models.

**Measurable Perspective.** The sandbox also provides many classes to encapsulate observational capabilities, enabling subsequent optimization. For example, Data-Juicer-based classes can efficiently compute metrics such as text perplexity, video aesthetic value, and image quality scores with GPT API calls, along with statistical values like mean, variance, and percentiles. For models, various evaluation benchmarks are supported, like VBench and FVD (Unterthiner et al., 2019) for synthesized video assessment, and TextVQA (Singh et al., 2019), MMBench (Liu et al., 2024b), and MME (Fu et al., 2023) for image-to-text evaluation.

More detailed developmental information can be found with the source code and accompanying documentation in our open-sourced link, which are also continuously maintained online and evolved. For example, users can conveniently run the following one-line entry script to easily run the sandbox and flexibly adjust configurations to switch between many built-in hooks and capabilities.

```
python tools/sandbox_starter.py \
      --config configs/demo/sandbox.yaml
```

### B.3. Extensibility of the Sandbox Suite

As a middleware, the sandbox itself does not impose any additional specific hardware dependencies. Instead, it inherits the dependencies of the integrated underlying libraries/frameworks and is continuously maintained to incorporate more community efforts in model training and evaluation. To simplify dependencies and avoid redundancy in an "all-in-one" environment, we have introduced and employed a lazy-loader mechanism at the Python package level. The code and parameters are provided and adjusted to ensure a stable and cost-controlled experiment environment, where, for example, a single GPU card can complete an end-to-end workflow within one day, enabling rapid feedback.

Besides, the capabilities of its underlying libraries are still evolving. For example, the utilized Data-Juicer's operators are not limited to data filters investigated on the main page; they now encompass a wide range of functionalities, including 100+ Mappers, Filters, Deduplicators, and Selectors. This diversity allows for research on various types of data processing utilities within the Sandbox. Since Mappers can be employed to examine the effects of data augmentation and editing on downstream model task performance, which we reserved further exploration beyond filters as future work.

## C. Analysis on the Costs of the Proposed Workflow

In this section, we give a more detailed discussion on the factors presented in the main page Sec. 3.2.4.

### C.1. Impact of Sampling Ratio $r$ for Data Pools

In this section, we give some theoretical discussion to demonstrate the rationality of why scaling data pools can work in a cost-effective way.

Following the notations defined in Sec. 3.2.4, the probability that the error introduced by our experiments on the small-scale data pools is greater than $\epsilon$ can be denoted as:

$$\mathbb{P}[\Delta_{pool} - \mathbb{E}[\Delta_{full}] \geq \epsilon].$$

It is worth noting that our sandbox aims to obtain effective feedback with minimal expenditure. This feedback is often less apparent than what is obtained from the full dataset and larger model, meaning that $\mathbb{E}[|\Delta_{full}|] \geq \mathbb{E}[|\Delta_{pool}|]$. For operators that have a positive effect, we have $\mathbb{E}[\Delta_{full}] \geq$

$\mathbb{E}[\Delta_{pool}]$. In this case, we can conclude that:

$$\mathbb{P}[\Delta_{pool} - \mathbb{E}[\Delta_{full}] \geq \epsilon] \leq \mathbb{P}[\Delta_{pool} - \mathbb{E}[\Delta_{pool}] \geq \epsilon]$$
$$\leq e^{-2\epsilon^2/(b-a)^2}.$$

Here we use Hoeffding's inequality with the assumption that $\Delta_{pool} \in [a, b]$. As the sample rate $r$ increases, the variance of the improvement across different training trials decreases, which is positively related to $(b - a)^2$, and thus the right term decreases. In conclusion, the probability of the error exceeding $\epsilon$ decreases exponentially as $r$ increases.

### C.2. Impact of the Number $m$ of Planned Small-Pool Experiments

In the proposed sandbox workflow as illustrated in Sec. 3.2, we choose to split each data pool after applying our studied Data-Juicer OPs into three buckets according to the ranks of their generated statistics. The number The number of buckets acts as a multiplying factor to determine the total number of the planned small-pool experiments $m$, and finally the total cost of our experiments. It reflects the trade-off between total cost, data intervention intensity (via operator stats), and the informativeness of model feedback (the metric changes $\Delta$ on interested tasks compared to the models trained on random sampled data pools). More buckets lead to greater statistical differences between buckets with different ranks (especially the first and last ones), strengthening attribution to data processing effectiveness. However, more buckets also reduce per-bucket data, increasing the risk of inadequate data for models to exhibit reasonable $\Delta$.

As a result, we do not aim for models to be "training done" or "converged" in the sandbox workflow experiments. Instead, we want to observe enlightening changes—positive or negative—in models after targeted data intervention versus random data sampling. In our early experiments, we tested bucket counts of [2,3,4,5] to evaluate whether a model trained on randomly sampled data could reasonably decrease loss after one epoch and show statistically significant changes on downstream tasks. Our findings indicate that three buckets are empirically good for the studied scenarios. Once determined, all controlled experiments are aligned to one complete epoch and matched to the random pool data size.

## D. Implementation Details of Sandbox Experiments

### D.1. Strategies to Combine OPs According to Correlations

In addition to assembling the OPs with the overall best performance, we also incorporate an analysis of inter-OP relationships into our recipe formulation process. Our workflow accommodates two strategies, with specific applications detailed in Sec. E.4:

- The first method involves computing Pearson correlation coefficients between the statistics generated by these OPs. Using a hierarchical clustering algorithm (Ward Jr, 1963), we group the OPs into $k$ clusters. From each cluster, we select the OP whose data pool yields the strongest model performance to form potential combinations.

- Alternatively, leveraging the outcomes of single-OP tests, we calculate Pearson correlation coefficients for each pair of dimensions within the evaluation metrics. Hierarchical clustering is again employed to categorize the metrics into $k$ classes. The top-performing OP from each class is chosen to create the combinations. This approach allows us to investigate whether these combinations lead to concurrent improvements or mutual inhibition across the evaluative metrics.

### D.2. Evaluation Metrics

In the paper, we mainly report overall performance as the relative changes over the baseline in terms of the average across all model metrics with normalization as follows:

$$\frac{\sum_i^N s_i/N - \sum_i^N s_i'/N}{\sum_i^N s_i'/N} = \frac{\sum_i^N (s_i - s_i')}{\sum_i^N s_i'}, \qquad (1)$$

where $N$ is the number of involved metrics, $s_i$ is the score of $i$-th model measurement metric, $s_i'$ is the corresponding score gained by the baseline model trained on randomly sampled data. Below are the specific evaluation metrics involved in this study.

**TextVQA, MMBench, MME.** These benchmarks serve as critical evaluators of MLLM's proficiency in understanding images. TextVQA (Singh et al., 2019) specifically targets the assessment of MLLMs' abilities to read and reason about textual content embedded within images. MMBench (Liu et al., 2024b), a vast multimodal benchmark, encompasses perception and reasoning skills through a plethora of multi-choice questions, numbering in the thousands. Additionally, a Chinese translation, MMBench-CN, is integrated for broader accessibility. MME (Fu et al., 2023) focuses on the perceptual and cognitive competencies of MLLMs, incorporating 14 finely categorized subtasks, each addressing Yes/No inquiries underpinned by meticulously crafted guidelines.

**VBench.** We engage with VBench (Huang et al., 2024), a holistic benchmark suite tailored for the rigorous evaluation of video generative models. It facilitates granular and objective assessment across a spectrum of dimensions, deconstructing the concept of "video generation quality" into 16 discrete metrics. Each metric is assessed using a carefully

curated suite of prompts, comprising 946 unique prompts, with the requirement to produce 5 videos per prompt.

Owing to the disparity in evaluation criteria and the inherent variability across different modalities, we discern that the magnitude of performance fluctuation in the image-to-text generation substantially exceeds that observed in the text-to-video generation. This discrepancy underscores again the need for nuanced data-model co-development in addressing the complexities inherent in each modality.

Across all experiments, results are reported as averages with standard deviations from 2 to 5 repetitions for the image-to-text and text-to-video tasks, respectively, due to their differing levels of variance.

**Metrics for Image-Text Pre-training task.** For the CLIP experiments, we adopt 40 distinct benchmark scores used in DataComp (Gadre et al., 2023), e.g., the image classification on 30 data subsets for diverse scenarios like ImageNet derivatives, the image-text retrieval on 3 data subsets like Flicker and MSCOCO, the fairness related classification task like Dollar Street and GeoDE. For more details about these tasks, please refer to the DataComp and their original papers.

### D.3. Descriptions of Studied Operators

The study involves 31 OPs from Data-Juicer (Chen et al., 2024a). Their corresponding statistics and detailed descriptions are provided in Table 4.

### D.4. Image-to-Text Generation

Our first task focuses on foundational image understanding ability, by experimenting on Mini-Gemini (MGM-2B), a state-of-the-art (SOTA) 2 billion parameter multimodal LLM (Li et al., 2024b). The training protocol for MGM-2B involves two stages: pretraining and fine-tuning. Our experimental focus lies in the pretraining phase, which seeks to harmonize visual and textual representations. We utilize the original pretraining dataset as our original dataset $\mathcal{D}$, consisting of approximately 1.2M instances. We set the size of $\mathcal{D}_{sample}$ as 200k. The single-OP data pools $\mathcal{D}_i$ and multi-OP data pools $\mathcal{D}_{\mathcal{S}}$ are capped at a maximum of 200k instances, ensuring consistency of data pool size. To match the down-sampling rate used during pretraining, the fine-tuning dataset is sampled into a 240k instance subset.

We first conduct single-OP experiments (Sec. 4.2) that encompasses 22 text-image relevant OPs from Data-Juicer, split evenly between text-only and image-related multimodal OPs. After the two-stage training, model evaluation is conducted on established benchmarks including TextVQA (Singh et al., 2019), MMBench (Liu et al., 2024b), and MME (Fu et al., 2023).

For multi-OP data pools (Sec. 4.3), we identify the top-3 highest-performing OPs from single-OP experiments and study their possible combinations. Additionally, we analyze the correlations among the 23 data statistics produced by 22 OPs capable of generating instance-level stats [2]. Employing a hierarchical clustering algorithm (Ward Jr, 1963), these OPs are grouped into three clusters based on correlation coefficients, with the highest-performing OP from each cluster selected for combination testing. To ensure a robust and fair comparison, we must acknowledge the constraints imposed by the limited data volume within the highest-tier data pool. As the number of combinations increases, the available dataset size diminishes. In particular, the size of $\mathcal{P}_{\mathcal{S}}$ was reduced from 200k samples to 159k samples during the Top-3 combination experiments. Similarly, in the cluster-wise combination experiments, the dataset size decreased from 200k samples to 126k samples.

Next, in Sec. 4.4, we explore the optimal OP combination based on previous experiments and adopt the methodology from Sec. 3.2.3 for comparative experiments on training with repeated data versus non-repeated data. Note that due to filtering, the final instance count decreases from 200k to approximately 159k after the OP combination. These data are then repeated in increments from double to eightfold, mirroring the size of the original pretraining set.

Collectively, all these experiments yield profound insights into image-to-text model training, data processing, and iteration strategies from a data-model co-development perspective, further verified in the larger-scale scenario in Sec. 4.5.

In terms of the model training details, we train the MGM-2B model from scratch with less training data (about 1/6 of the original training datasets) in baseline experiments to make sure each experiment can be finished within one day. We keep every training setting (e.g. learning rate scheduler, global batch size) the same as the original model except for training datasets and training devices. For single-OP and OP combination experiments are trained on only 1 A100 GPU for each experiment so we increase the number of gradient accumulation steps from 4 to 32 to keep the same global batch size. For experiments of duplicating high-quality datasets, 8 A100 GPUs are involved to train the model, and the number of gradient accumulation steps is restored to 4. Each experiment is repeated 3 times with different random seeds to make the final results more comprehensive.

### D.5. Text-to-Video Generation

For the second task, text-to-video generation, we adopt the advanced DiT-based models, EasyAnimate (Xu et al., 2024b), which originally integrates diverse datasets totaling 1.2M instances from InternVid (Wang et al., 2023) (606k),

---

[2]The image height and width are produced by one OP.

Table 4: Overview of involved OPs in the study, including the modality they pertain to, along with their statistical data and detailed descriptions of these statistics.

| OP Name | Modality | Statistics | Description |
|---|---|---|---|
| `alphanumeric_filter` | text | *Alphanumeric Ratio* | Alphanumeric ratio in the text. |
| `character_repetition_filter` | text | *Character Repetition Ratio* | Char-level n-gram repetition ratio in text. |
| `flagged_words_filter` | text | *Flagged Word Ratio* | Flagged-word ratio in the text |
| `image_aesthetics_filter` | image | *Image Aesthetics Score* | Aesthetics score of the image |
| `image_aspect_ratio_filter` | image | *Image Aspect Ratio* | Aspect ratio of the image |
| `image_nsfw_filter` | image | *Image NSFW Score* | NSFW score of the image |
| `image_shape_filter` | image | *Image Width/Height* | Width and height of the image |
| `image_size_filter` | image | *Image Size* | Size in bytes of the image |
| `image_text_matching_filter` | text, image | *BLIP Image-Text Similarity* | Image-text classification matching score based on a BLIP model |
| `image_text_similarity_filter` | text, image | *CLIP Image-Text Similarity* | Image-text feature cosine similarity based on a CLIP model |
| `image_watermark_filter` | image | *Image Watermark Score* | Predicted watermark score of the image based on an image classification model |
| `language_id_score_filter` | text | *Language Score* | Predicted confidence score of the specified language |
| `perplexity_filter` | text | *Text Perplexity* | Perplexity score of the text |
| `phrase_grounding_recall_filter` | text, image | *Phrase Grounding Recall* | Locating recall of phrases extracted from text in the image |
| `special_characters_filter` | text | *Special Character Ratio* | Special character ratio in the text |
| `stopwords_filter` | text | *Stopword Ratio* | Stopword ratio in the text |
| `text_action_filter` | text | Text Action Number | Number of actions in the text |
| `text_entity_dependency_filter` | text | *Entity Dependency Number* | Number of dependency edges for an entity in the dependency tree of the text |
| `text_length_filter` | text | *Text Length* | Length of the text |
| `token_num_filter` | text | *Token Number* | Token number of the text |
| `video_aesthetics_filter` | video | *Video Aesthetics Score* | Aesthetics score of sampled frames in the video |
| `video_aspect_ratio_filter` | video | *Video Aspect Ratio* | Aspect ratio of the video |
| `video_duration_filter` | video | *Video Duration* | Duration of the video |
| `video_frames_text_similarity_filter` | text, video | *Frames-Text Similarity* | Similarities between sampled frames and text based on a CLIP/BLIP model |
| `video_motion_score_filter` | video | *Video Motion Score* | Motion score of the video |
| `video_nsfw_filter` | video | *Video NSFW Score* | NSFW score of the video |
| `video_ocr_area_ratio_filter` | video | *Video OCR-Area Ratio* | Detected text area ratio for sampled frames in the video |
| `video_resolution_filter` | video | *Video Width/Height* | Width and height of the video |
| `video_watermark_filter` | video | *Video Watermark Score* | Predicted watermark score of the sampled frames in the video based on an image classification model |
| `words_num_filter` | text | *Word Number* | Number of words in the text |
| `word_repetition_filter` | text | *Word Repetition Ratio* | Word-level n-gram repetition ratio in the text |

Panda-70M (Chen et al., 2024e) (605k), and MSR-VTT (Xu et al., 2016) (6k). The studied baseline model is trained on a subset of 40k instances, employing LoRA (Hu et al., 2022) for efficiency. As a result, the size of $\mathcal{D}$ is 1.2M, and the size of $\mathcal{D}_{sample}$, the single-OP data pools $\mathcal{D}_i$ and multi-OP data pools $\mathcal{D}_{\mathcal{S}}$ are all 40k. Model outputs are assessed using VBench (Huang et al., 2024) across 16 metrics on video quality and video-text matchness.

Our investigation covered 21 OPs, including 13 text-only OPs and 10 video-related multimodal OPs. Analogous to the image-to-text generation, we conduct single-OP and multi-OP combination experiments, in Sec. 4.2 and Sec. 4.3 respectively. However, given the reduced relevance of data statistics in video-related OPs, our analysis centers on the correlations among the 16 VBench evaluation metrics. These metrics are clustered into three groups, with the best-performing OP selected from each group.

Through OP combination experiments, we pinpoint the most effective set of OPs. In Sec. 4.4, we then sample 40k instances from the filtered data pool and repeat the training process for up to 10 epochs. For comparative analysis, we adhere to the method outlined in Sec. 3.2.3, selecting larger data volumes (80k, 120k, ..., up to 400k instances) for single-epoch training. To examine the effectiveness of the derived insights for text-to-video data-model co-development, we finally incorporate them into larger-scale scenarios in Sec. 4.5.

In terms of the model training details, we adopt the advanced DiT-based EasyAnimate (Xu et al., 2024b) model, which integrates diverse datasets totaling 1.2M instances from InternVid (Wang et al., 2023) (606k), Panda-70M (Chen et al., 2024e) (605k), and MSR-VTT (Xu et al., 2016) (6k). Baseline experiments are executed on a subset of 40k instances, employing LoRA (Hu et al., 2022) for efficiency. During training, we maintain a video resolution of 256x256, sample every other frame, and randomly select sequences of 16 consecutive frames. The training process involves performing a backward pass for the loss of every 8 samples, with single-OP and OP combination experiments trained on a single GPU with a batch size of 8 for 5k steps, amounting to approximately 16 GPU hours per training run. Experiments for duplicating high-quality data, as well as larger-scale training, are conducted with a batch size of 1 across 8 GPUs. The models employ the Adam optimizer for training, with a learning rate set to $2 \times 10^{-5}$, weight decay parameter at $3 \times 10^{-2}$, and epsilon configured to $10^{-10}$. Each experiment is repeated twice with random seeds of 42 and 45, respectively.

### D.6. Image-Text Pre-training

For the third task, image-text pre-training, we adopt the well-studied CLIP model (Radford et al., 2021). Specifi-

cally, we utilize data from the small track of the DataComp competition (Gadre et al., 2023) and adhere to its evaluation metrics, which include 40 distinct evaluation subsets. Due to some broken links, we successfully downloaded 85.2% of the dataset, resulting in a total of 10.9 million samples as our $\mathcal{D}$. All baseline models were trained on an equivalent volume of data as used in the contrastive experiments, sampled randomly from this dataset.

In the multi-OP experiments (see Fig. 2(c)), due to data limitations in the top-level data pool, the training dataset size is reduced to 880k for the top recipes. Correspondingly, the default computation scale of DataComp is adjusted to 0.25 expansion rate for the subsequent compute scaling experiments (see Fig. 3(c)).

## E. Additional Experimental Results

### E.1. Summarization of Sandbox Experiments varying Multiple Scales

The Table 5 provides an overview of our sandbox experiments on image-to-text generation, text-to-video generation, and image-text pre-training tasks.

### E.2. Complete Single-Operator Ranking

In Table 6, Table 7, and Table 8, we present complete numeric results conducted on individual OP experiments (Sec. 4.2), from which we can discern some more detailed observations.

In image-to-text generation, it is preferable for the input of training images to align as closely as possible with the original configuration of the vision tower, such as training dimensions (height, width, and sizes). Additionally, CLIP similarity scores tend to be more influential than BLIP similarity scores. The BLIP similarity does not show much distinction and paradoxically, a lower similarity often results in better performance, which defies common sense. Images with excessively high aesthetic quality may offer limited assistance in feature alignment, while watermarks might have certain impacts on the OCR performance of the model.

In text-to-video generation, having a consistent aspect ratio for the training data is better than having ratios that are inconsistent but close to the 1:1 ratio used during training. For instance, a data pool with a 'middle' video aspect ratio consistently at 16:9 performs optimally. Videos with high video aesthetics scores and low video NSFW scores, as well as those with low video OCR-area ratios and high video motion scores, tend to be of higher quality. While single-text-related operators might not be as critical in text-to-video generation, they can still effectively filter out some dirty data.

Table 5: Overview of sandbox scenarios and their effective evidence across different scales.

| Sandbox Scenario | Image-to-Text Generation | Text-to-Video Generation | Image-Text Pretraining | Image Captioning Fine-tuning |
|---|---|---|---|---|
| **Main Effectiveness Evidence** | Optimal recipe derived from small data pools achieves superior model performance in larger data pools, using only half of baseline's compute cost. | Insights from small data pools results in VBench-Top1 model, transferring across data size, model scales, and model architectures. | Best recipe identified in the model with fewer FLOPS maintains optimal performance with increased model FLOPS and compute resources, showing clear scaling behaviors in power-law form. | Optimal recipe identified with InternVL-2.0 model, leads to consistent and steady performance advantages as the model scale increases from 1B to 26B. |
| **Model Scale Range** | MGM-2B in all experiments | EasyAnimate to T2V-Turbo (Heterogeneous architectures) | CLIP: ViT-B-32 to ViT-B-16 (Different FLOPS) | InternVL-2 with 1B, 2B, 4B, 26B parameters |
| **Data Scale Range (w.r.t Distinct Dataset Size)** | 126k to 200k | 40k to 147k and 228k | 880k to 2,683k | 24k to 189k |
| **Compute Scale Range (w.r.t Number of Trained Sample)** | 1 to 8 Epochs | 1 to 10 Epochs | 4 to 14 Epochs | 1 epoch in all experiments |

Table 6: The complete OP ranking, including their statistical dimensions and the improvements relative to the baseline. We consider three splits with low, middle, and high statistical values for each OP. The baseline used is based on random sampling with equal data volume.

| Task | OP-Generated Statistics | Average Performance Changes (%) | | |
|------|-------------------------|------------------|-----------------|------------------|
| | | Data Pool (Low) | Data Pool (Mid) | Data Pool (High) |
| | *Image NSFW Score* | 7.13 ± 4.29 | 18.44 ± 18.45 | **66.38 ± 32.65** |
| | *Text Action Number* | **59.90 ± 46.49** | 0.29 ± 2.16 | -2.05 ± 2.48 |
| | *Language Score* | **49.90 ± 53.82** | 0.85 ± 2.87 | -1.43 ± 2.40 |
| | *CLIP Image-Text Similarity* | 1.20 ± 4.86 | -1.81 ± 2.88 | **49.81 ± 44.72** |
| | *Phrase Grounding Recall* | -0.49 ± 3.87 | -0.58 ± 6.12 | **49.39 ± 29.83** |
| | *Image Width* | **42.04 ± 57.27** | 10.31 ± 12.59 | 1.35 ± 4.36 |
| | *Special Character Ratio* | -3.08 ± 0.63 | -0.75 ± 1.61 | **39.67 ± 58.82** |
| | *Flagged Word Ratio* | **38.48 ± 27.76** | -0.39 ± 0.43 | 22.49 ± 29.81 |
| | *Image Height* | **35.66 ± 48.62** | 12.91 ± 10.42 | 18.73 ± 27.32 |
| | *Word Repetition Ratio* | **33.14 ± 23.39** | 2.59 ± 5.31 | -0.55 ± 2.90 |
| | *Text Length* | **30.67 ± 28.54** | -0.44 ± 0.73 | -3.71 ± 0.39 |
| Image-to-Text | *Stopword Ratio* | 3.34 ± 5.05 | **24.62 ± 36.73** | -1.56 ± 1.59 |
| | *Image Size* | 0.76 ± 0.55 | **19.16 ± 27.29** | 1.58 ± 2.20 |
| | *Text Perplexity* | -1.69 ± 1.30 | 16.70 ± 24.49 | **18.26 ± 23.02** |
| | *Image Aesthetics Score* | 11.94 ± 12.21 | **16.58 ± 25.70** | 0.16 ± 3.67 |
| | *Word Number* | **15.96 ± 29.01** | -2.48 ± 0.26 | -1.97 ± 2.05 |
| | *BLIP Image-Text Similarity* | **11.76 ± 22.83** | 1.74 ± 2.49 | 1.34 ± 2.21 |
| | *Image Watermark Score* | -1.50 ± 2.41 | 7.51 ± 12.82 | **11.54 ± 13.14** |
| | *Alphanumeric Ratio* | 2.35 ± 7.63 | -0.66 ± 0.69 | **8.71 ± 12.87** |
| | *Character Repetition Ratio* | 0.00 ± 1.13 | -1.42 ± 0.60 | **7.94 ± 14.63** |
| | *Entity Dependency Number* | 1.35 ± 1.81 | -0.87 ± 1.15 | **6.67 ± 8.44** |
| | *Token Number* | **6.31 ± 7.86** | 0.80 ± 0.92 | 0.33 ± 6.45 |
| | *Image Aspect Ratio* | 0.00 ± 1.34 | **1.89 ± 2.71** | -0.02 ± 1.12 |
| | *Video Aesthetics Score* | -0.98 ± 0.08 | 0.13 ± 0.09 | **0.96 ± 0.13** |
| | *Video NSFW Score* | **0.82 ± 0.36** | -0.05 ± 0.07 | -0.57 ± 0.07 |
| | *Frames-Text Similarity* | -1.45 ± 0.69 | 0.23 ± 0.21 | **0.79 ± 0.15** |
| | *Special-Characters Ratio* | **0.54 ± 0.36** | -0.13 ± 0.70 | -0.14 ± 0.10 |
| | *Token Number* | **0.53 ± 0.04** | 0.18 ± 0.32 | 0.41 ± 0.25 |
| | *Character Repetition Ratio* | -0.29 ± 0.27 | **0.47 ± 0.80** | 0.18 ± -0.52 |
| | *Video Height* | -0.10 ± 0.21 | 0.12 ± 0.13 | **0.46 ± 0.44** |
| | *Video OCR-Area Ratio* | **0.44 ± 0.04** | 0.02 ± 0.63 | -0.66 ± 0.23 |
| | *Word Number* | -0.49 ± 0.07 | -0.41 ± 0.72 | **0.44 ± 0.45** |
| | *Entity Dependency Number* | **0.40 ± 0.01** | 0.28 ± 0.48 | -0.18 ± 0.44 |
| Text-to-Video | *Text Action Number* | 0.18 ± 0.56 | -0.71 ± 0.28 | **0.37 ± 0.28** |
| | *Alphanumeric Ratio* | -0.10 ± 0.19 | 0.20 ± 0.19 | **0.33 ± 0.17** |
| | *Video Motion Score* | -0.55 ± 0.40 | **0.33 ± 0.21** | 0.32 ± 0.15 |
| | *Video Watermark Score* | -0.27 ± 0.27 | -0.25 ± 0.25 | **0.29 ± 0.16** |
| | *Text Perplexity* | **0.15 ± 0.69** | -0.13 ± 0.27 | 0.09 ± 0.56 |
| | *Stopword Ratio* | -0.01 ± 0.05 | -0.48 ± 0.22 | **0.12 ± 0.07** |
| | *Video Aspect Ratio* | -0.32 ± 0.14 | **0.11 ± 0.18** | -0.02 ± 0.40 |
| | *Language Score* | -0.21 ± 0.01 | -0.03 ± 0.38 | **0.09 ± 0.03** |
| | *Word Repetition Ratio* | 0.00 ± 0.17 | **0.06 ± 0.24** | -0.43 ± 0.24 |
| | *Video Duration* | -0.58 ± 0.05 | -0.16 ± 0.09 | **0.04 ± 0.84** |
| | *Text Length* | -0.09 ± 0.63 | -0.66 ± 0.08 | **0.03 ± 0.22** |

Table 7: The complete OP ranking, including their statistical dimensions and the improvements relative to the baseline for image-text similarity/classification task. We consider three splits with low, middle, and high statistical values for each OP. The baseline used is based on random sampling with equal data volume.

| OP-Generated Statistics | Average Performance Changes (%) | | |
|---|---|---|---|
| | Data Pool (Low) | Data Pool (Mid) | Data Pool (High) |
| *CLIP Image-Text Similarity* | -32.57 | -6.39 | **39.53** |
| *BLIP Image-Text Similarity* | -24.28 | 1.82 | **25.39** |
| *Image NSFW Score* | **12.18** | 1.28 | -18.38 |
| *Word Number* | -18.65 | 0.74 | **9.78** |
| *Stopword Ratio* | -4.28 | -3.33 | **8.97** |
| *Special Character Ratio* | **8.86** | 4.15 | -16.03 |
| *Phrase Grounding Recall* | **7.79** | 1.85 | -10.60 |
| *Text Length* | -8.31 | 1.81 | **7.29** |
| *Character Repetition Ratio* | 1.99 | 0.04 | **6.63** |
| *Image Aspect Ratio* | 4.93 | -4.55 | **5.87** |
| *Text Perplexity* | **5.27** | 2.46 | -9.56 |
| *Image Width* | -6.66 | 4.97 | **5.23** |
| *Image Height* | -4.03 | **5.02** | 0.89 |
| *Image Size* | -12.11 | **5.00** | 2.87 |
| *Image Aesthetics Score* | -9.61 | -8.13 | **4.64** |
| *Image Watermark Score* | **3.84** | -3.74 | -4.72 |
| *Flagged Word Ratio* | **3.66** | 3.47 | 1.59 |
| *Entity Dependency Number* | -5.53 | -0.39 | **2.50** |
| *Word Repetition Ratio* | -3.16 | -0.91 | **1.84** |
| *Alphanumeric Ratio* | -2.55 | **1.65** | 0.63 |
| *Token Number* | -6.35 | **1.44** | 0.27 |

In image-text pre-training, we can see that the top three performing operations are the ones generating stats of *CLIP Image-Text Similarity*, *BLIP Image-Text Similarity*, and *Image NSFW Score*. The first two OPs generate statistics based on auxiliary CLIP and BLIP models, enlightening that the modality alignment degree is critical to the studied pair-wise learning task.

In image captioning fine-tuning, models trained on shorter captions (low range of *Text Length*) achieve the best performance. Captions with less repetition on both character and word levels help to train models with better performance. It suggests that in the image captioning task, the quality of captions is the key to better performance.

### E.3. Diversity Analysis

In this subsection, we delve into the diversity of the data in interested data pools. From the perspective of quantitative indicators, we confine our focus to statistical analysis of words within text data and compute their entropy. We operate under the assumption that the texts provide an accurate description of the images and videos. Consequently, the diversity inferred from the texts also serves as a proxy for the diversity of the associated images and videos.

The Table 9 shows the entropy of text words for different

data pools, which can be normalized as follows:

$$\sum_w -\mathcal{P}(w) \log \mathcal{P}(w), \quad (2)$$

where $w$ is a word and $\mathcal{P}$ is the distribution of words in a data pool. As we can see, data pools with higher *NSFW scores* and lower *Language Score* have higher word entropy, suggesting greater diversity within these data pools.

We also introduce a more direct way to show the data diversity to users. We use a tagging model to extract core objects, concepts, and domains from the contents of the datasets, including both texts and multimodal content. Then we analyze the domain-specific distribution via word clouds of these tags. We show an example of analyzed word clouds from videos on different data pools of *Video NSFW Score* in Fig. 5. As we can see, although the data pool with high video NSFW scores is the most diverse one, our model achieves the best performance on the data pool with low video NSFW scores, which is consistent with the argument presented in Sec. 4.2.

### E.4. Correlation Analysis

To investigate the intrinsic relationships between OPs and to support our recipe formulation, we explore relevance from the following two perspectives.

Table 8: The complete OP ranking, including their statistical dimensions and the improvements relative to the baseline for the image captioning task. We consider three splits with low, middle, and high statistical values for each OP. The baseline used is based on random sampling with equal data volume.

| OP-Generated Statistics | Average Performance Changes (%) | | |
|---|---|---|---|
| | Data Pool (Low) | Data Pool (Mid) | Data Pool (High) |
| *Text Length* | **0.760** | -3.125 | -11.364 |
| *Image Watermark Score* | -0.637 | -0.132 | **0.477** |
| *Character Repetition Ratio* | **0.449** | -0.461 | -0.634 |
| *Text Perplexity* | -1.505 | **0.331** | -3.220 |
| *Word Repetition Ratio* | **0.300** | -3.125 | -11.349 |
| *Image Width* | **0.265** | -0.230 | -0.112 |
| *CLIP Image-Text Similarity* | -0.920 | **0.225** | -2.101 |
| *Phrase Grounding Recall* | -1.318 | 0.166 | **0.188** |
| *Image Aesthetics Score* | -0.300 | **0.186** | -0.243 |
| *Image Aspect Ratio* | **0.171** | -0.490 | -0.118 |
| *Image NSFW Score* | -0.030 | 0.062 | 0.113 |
| *Image Height* | -0.080 | -0.030 | **0.088** |
| *Stopword Ratio* | -3.795 | 0.084 | -4.930 |
| *Image Size* | -0.112 | -0.071 | **0.028** |
| *Flagged Word Ratio* | -0.012 | -0.663 | -0.429 |
| *Language Score* | -0.154 | -1.534 | -7.823 |
| *Token Number* | -0.610 | -0.987 | -9.017 |
| *Alphanumeric Ratio* | -0.795 | -4.300 | -11.592 |
| *Text Action Number* | -2.633 | -0.931 | -8.565 |
| *Word Number* | -1.125 | -2.078 | -9.333 |
| *Special Character Ratio* | -11.194 | -4.782 | -1.145 |
| *Entity Dependency Number* | -1.443 | -1.660 | -7.072 |
| *BLIP Image-Text Similarity* | -2.254 | -1.446 | -6.221 |

Table 9: The entropy of text words for data pools with different levels of *Image NSFW score* and *Language score*.

| Task | OP-Generated Statistics | Word Entropy | | |
|---|---|---|---|---|
| | | Data Pool (Low) | Data Pool (Mid) | Data Pool (High) |
| Image-to-Text | *Image NSFW Score* | 6.97 | **7.35** | 7.29 |
| | *Language Score* | **7.47** | 7.32 | 6.98 |
| Text-to-Video | *Video NSFW Score* | 5.84 | **6.03** | 6.01 |
| | *Language Score* | **6.30** | 5.85 | 5.73 |

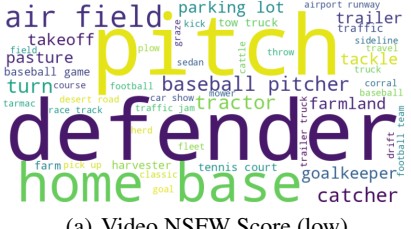

(a) Video NSFW Score (low)

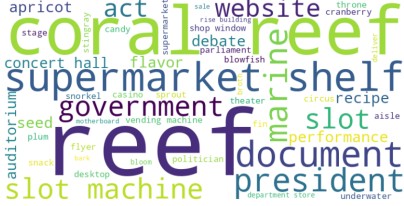

(b) Video NSFW Score (mid)

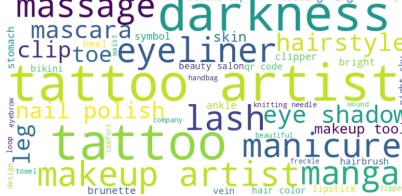

(c) Video NSFW Score (high)

Figure 5: Domain-specific word clouds tagged from videos for data pools with different levels of and *Video NSFW score*.

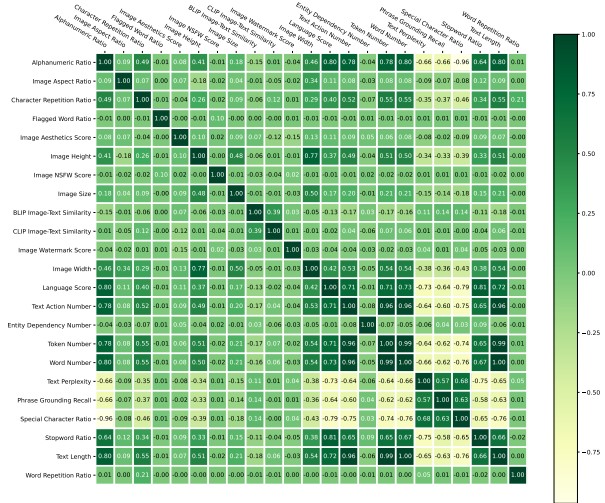

Figure 6: Pearson correlation coefficients for OP statistics in image-to-text generation.

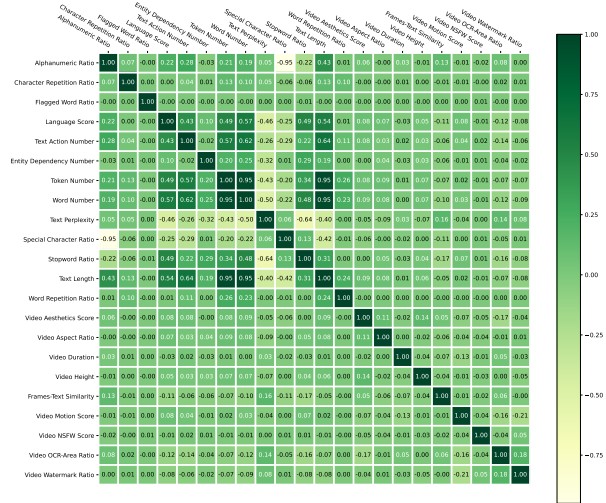

Figure 7: Pearson correlation coefficients for OP statistics in text-to-video generation.

First, we adopt the most direct approach by examining the Pearson correlation coefficients between the statistics of OPs, as illustrated in Figures 6, 7, 8, and 9. Intuitively, the associations between the statistics of OPs utilized in image-to-text generation appear to be significantly stronger than those in text-to-video generation. For instance, in image-to-text generation, phrase grounding recall shows a strong positive correlation with text perplexity and special character ratio, while exhibiting a strong negative correlation with the alphanumeric ratio, language score, number of text actions, stopword ratio, and text length. In contrast, in text-to-video generation, we observe relationships primarily among the purely textual OPs, while video-related operators are largely orthogonal to others. Therefore, for image-to-text generation, we categorize OPs into three groups based on the correlation of their statistics and select the optimal OP from each category to create combinations. However, this method does not appear appropriate for text-to-video generation due to the sparser correlations observed.

In the second approach, we categorize the evaluation metrics based on their correlations, as illustrated in Fig. 10 for image-to-text generation and Fig. 11 for text-to-video generation. In image-to-text generation, several evaluation metrics are highly specific and possess unique characteristics, such as those for OCR and coding tasks, resulting in a greater number of categories upon classification. In contrast, VBench's evaluation metrics can be broadly divided into several classes, including static video metrics (e.g., subject and background consistency), dynamic metrics (e.g., dynamic degree), and video quality indicators (e.g., aesthetic quality and imaging quality).

Notably, the dynamic degree negatively correlates with many other metrics, particularly those favoring static videos,

thus preventing videos without movement from being rated as optimal. Based on these observations, for text-to-video generation, we apply a hierarchical clustering algorithm (Ward Jr, 1963) to classify the VBench metrics into three categories based on their correlations: static video metrics, dynamic video metrics, and video quality along with video-text matching. We then select the best-performing OP for each of these three metric categories, where each excels in different aspects. These selected OPs are subsequently combined and experimentally evaluated together.

### E.5. Recipes Based on Correlation Analysis

In addition to selecting the overall best-performing OPs, we aim to identify operators with distinct advantages to explore whether combining these operators can synergistically leverage their strengths for improved outcomes. We selected operators with unique strengths based on **correlation information** obtained from single-operator experiments. Detailed correlation analyses can be found in Appendix E.4.

Specifically, for the image-to-text generation, we categorize the OPs by calculating correlations between their statistics. We then select representative operators within each category: `TextActionFilter` for text, `ImageNSFWFilter` for images, and `PhraseGroundingRecallFilter` for combined text-image relevance. For the text-to-video generation, based on single-operator experiments, we categorize the metrics of VBench (Huang et al., 2024) into three classes according to their relevance and select the best-performing operator from each category. The chosen operators are `VideoMotionScoreFilter`, which excels in video static features; `VideoDurationFilter`, which is superior in dynamic features; and

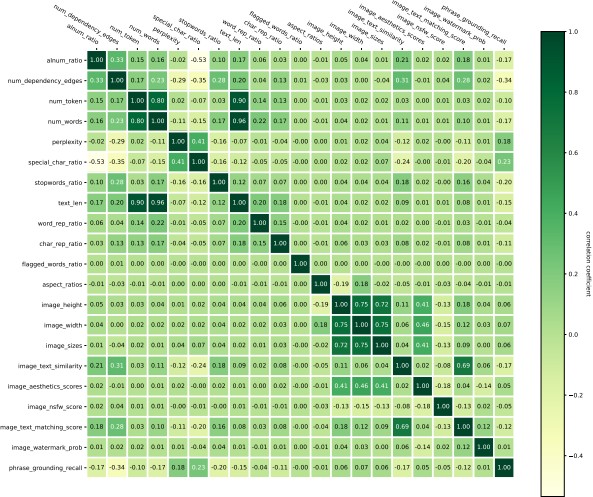

Figure 8: Pearson correlation coefficients for OP statistics in image-text pre-training.

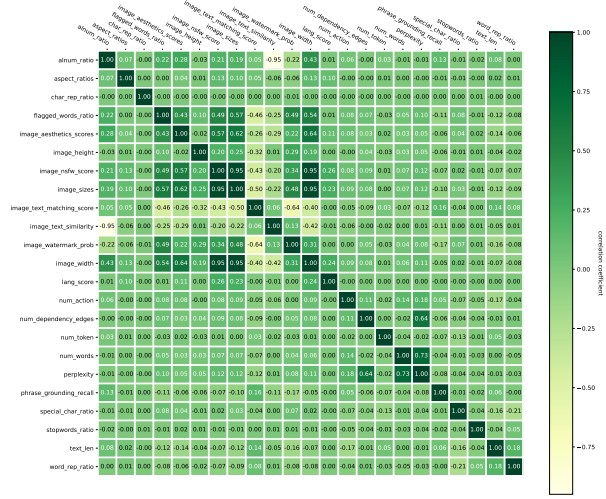

Figure 9: Pearson correlation coefficients for OP statistics in image-captioning oriented fine-tuning.

`VideoAestheticsFilter`, which exhibits the best composite performance in video quality and video-text matching. We summarize the experiment results in Figures 12(a) and 12(b).

> **Observation 8 (Effect of Orthogonal Combination)**
>
> Combining OPs that excel in orthogonal dimensions on model or data does not guarantee complementary effects; rather, it is more likely that they will impede each other's performance.

As depicted in Figures 12(a) and 12(b), regardless of how these top-performing OPs are combined, they ultimately reduce the model's performance in both image-to-text and text-to-video generation. This observation challenges the naive assumption widely used in existing SOTA works, that various intuitively useful data cleansing actions, when stacked serially, can synergistically enhance performance.

### E.6. Detailed Evaluation Results on MGM

The detailed experimental results in terms of different evaluation benchmarks are shown in Table 10.

### E.7. Ablation Study Based on T2V-Turbo

The results are shown in Table 3 representing the enhancements we achieved based on T2V-Turbo (Li et al., 2024a). T2V-Turbo applies LoRA (Hu et al., 2022) to VideoCrafter-2.0 (Chen et al., 2024d) and is trained on the WebVid (Bain et al., 2021) dataset, using VideoCrafter-2.0 as a teacher for distillation and incorporating reinforcement learning with rewards for the generated videos. The loss $L$ of T2V-Turbo

is defined as:

$$L = L_{CD} - 1 \times R_{img} - 2 \times R_{vid}, \quad (3)$$

where $L_{CD}$ is the loss of consistency distillation (Song et al., 2023), $R_{img}$ is the reward of image quality of generated video frames from HPSv2.1 (Wu et al., 2023) and $R_{vid}$ is the reward of video quality from InternVideo2 (Intern-Vid2 S2) (Wang et al., 2024b). In the paper, we make the following cumulative modifications to T2V-Turbo:

1. **Proposed Data Pool (Enhancement from Data View).** Utilizing the optimal data pool identified in Sec. 4.4, which consists of 147k instances with low video NSFW scores and high frame-text similarities, we replace the WebVid data and train for 6 epochs based on the insight from Sec. 4.4.

2. **Initialization for LoRA (Enhancement from Model View).** We use the T2V-Turbo model to initialize the parameters for training with LoRA.

3. **Self-Distillation. (Enhancement from Model View).** When we initialize with the T2V-Turbo model, the student model is already outperforming the teacher model, VideoCrafter-2.0, potentially leading to unstable training. To mitigate this, we use the T2V-Turbo model itself as the teacher to ensure that the reinforcement learning does not diverge excessively.

4. **Real-Data Loss (Data-Model Co-Enhancement).** To enhance the role of the proposed data pool during training, we add a real-data loss between the generated videos and the input videos to the distillation loss and reward loss. Furthermore, we set the weights of both the real-data loss and the distillation loss to 0.5. The modified

Table 10: Performance of MGM-2B with different pretraining datasets. MGM-2B pretrained with only 1/10 distinct instances and 1/2 of the total instances performs better. than the baseline trained on the full dataset. The "*" indicates our reproduced version that is comparable with the official version.

| MGM-2B | Num. of Instances | Avg. Perf. Changes (%) | MMBench | MMBench-CN | MME-Perception | MME-Cognition |
|---|---|---|---|---|---|---|
| Baseline* | 1226k | - | 59.2 | 51.6 | 1334 | 302 |
| Data-Juicer | **159k (x4)** | **+2.12** | 62.08 | 52.32 | 1323.37 | 311.07 |

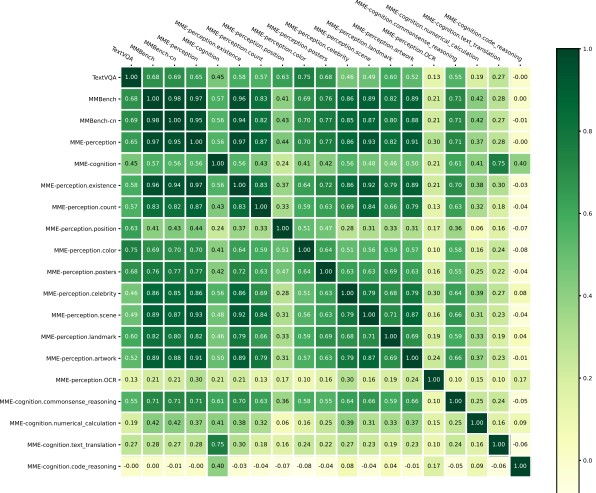

Figure 10: Pearson correlation coefficients for each dimension in TextVQA, MMBench, and MME.

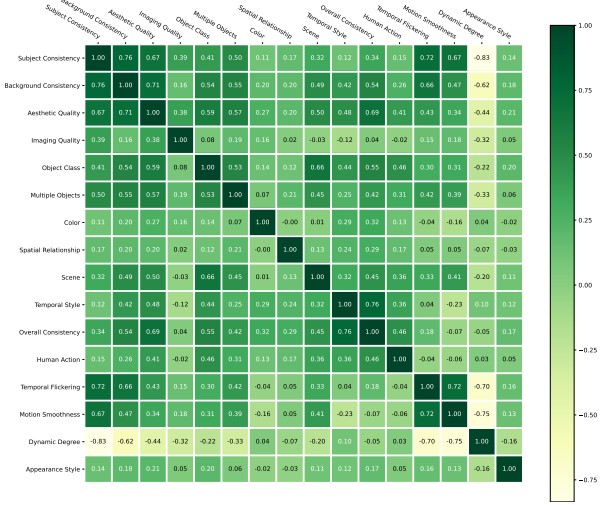

Figure 11: Pearson correlation coefficients for evaluation metrics in VBench.

loss from Equation 3 can be specified as:

$$L = 0.5 \times L_{CD} + 0.5 \times L_{real} - 1 \times R_{img} - 2 \times R_{vid}, \quad (4)$$

where $\times L_{real}$ is the real-data loss. Training on 147k instances with 6 epochs, we obtain the model, *Data-Juicer (T2V, 147k)*.

5. **Self-Evolution (Data-Model Co-Enhancement).** To validate the scalability of our methodologies, we expanded the original data pool and, by applying the `video_aesthetics_filter` and `video_motion_score_filter` to the existing recipes, we select an additional 228k instances to evolutionary self-distill *Data-Juicer (T2V, 147k)*. Meanwhile, in order to further amplify the impact of data, we have reduced the emphasis on reinforcement learning, thereby:

$$L = 0.5 \times L_{CD} + 0.5 \times L_{real} - 0.2 \times R_{img} - 0.4 \times R_{vid}. \quad (5)$$

We finally get *Data-Juicer (DJ, 228k)* in 4 epochs training.

Table 11 presents the ablation experiments of our modifications in the VBench evaluation. It is clear that when we replace the WebVid data with our proposed data pool, the model experiences a notable improvement, with the total score increasing from 81.01% to 81.84%. Subsequently, initializing the LoRA training parameters with the T2V-Turbo model does not lead to further enhancements in model performance. We suspect this might be because the teacher model is less effective than the T2V-Turbo model. Therefore, we use the T2V-Turbo model for self-distillation. While this method effectively raises the semantic score, it results in unstable video generation characterized by significant temporal flickering, which severely lowers the video quality. To counteract this, we add a real-data loss with the input data to secure the quality of the generated videos. Moreover, we evolve the model by using our trained model as the teacher model for continuous training on additional data, rather than T2V-Turbo. Ultimately, as we continue to enhance both the quality and semantic scores, we establish a new state-of-the-art.

### E.8. Full Results on VBench leaderboard

Backed by our proposed methodology and experiments, our Data-Juicer (DJ, with 228k) model refreshes the state-of-the-art on VBench (Huang et al., 2024), surpassing models

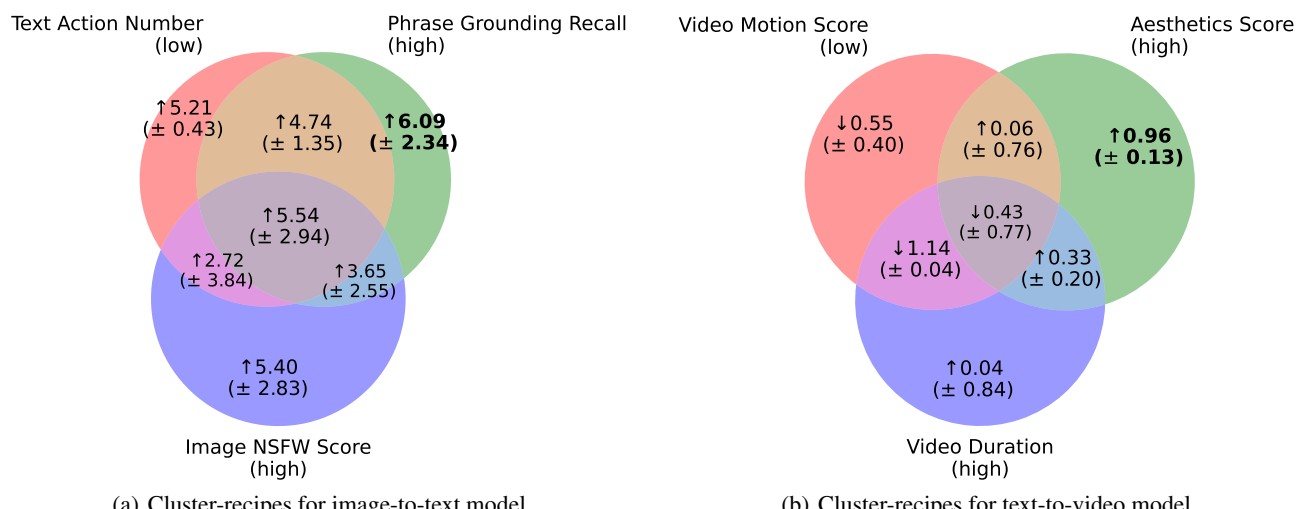

(a) Cluster-recipes for image-to-text model  (b) Cluster-recipes for text-to-video model

Figure 12: The improvements (%) of recipes with different OP combinations. The cluster recipes stem from the combinations of the best OPs in 3 categories. In the image-to-text generation, due to the limited data volume in the top-level data pool, the size of $\mathcal{P}_S$ decreased from 200k to 126k in the cluster recipes.

Table 11: Our model undergoes ablation experiments on the VBench leaderboard evaluation. Each '+' sign in the row indicates that the modification is added on top of the previous row's configuration.

| Model | Total Score (%) | Quality Score (%) | Semantic Score (%) |
|---|---|---|---|
| T2V-Turbo (VC2) | 81.01 | 82.57 | 74.76 |
| + Enhanced Data Pool | 81.84 | 83.40 | 75.60 |
| + Initialization for LoRA | 81.82 | **83.47** | 75.19 |
| + Self-Distillation | 79.16 | 79.48 | 77.92 |
| + Real-Data Loss | 82.10 | 83.14 | 77.93 |
| + Self-Evolution | **82.53** | 83.38 | **79.13** |

like Gen-3 (RunwayML, 2024) and Kling (Kuaishou, 2024), as illustrated in Fig. 13 and detailed in Table 3.

### E.9. Cost Comparison of all Sandbox Experiments

Besides, we present the compute details for our experiments and compare them to other baselines. Our experiment on a single data pool is designed to be completed within one day using a GPU. Consequently, with 8 A100 GPUs, we can run all our experiments within approximately 3 days, 7 days, and 10 days for the CLIP, image-to-text, and text-to-video experiments, respectively. The training sample size is in the order of $O(1)$ million video-text pairs and $O(1)$ million image-text pairs in our experiments. As a reference, the pre-training cost for a cutting-edge text-to-video model like Meta's MovieGen involves 6,144 H100 GPUs, $O(100)$ million video-text pairs, and $O(1)$ billion image-text pairs.

More specifically, for the image-to-text experiments, in Table 2, our MGM model shows a +2.12% average performance improvement over baseline, using approximately

51.8% of the FLOPS.

For the CLIP experiments, refer to Figure 12; the FLOPs with full expansion rate are 189 $\alpha$ PFLOPS (14.78G inference FLOPS per sample) and 526 $\alpha$ PFLOPS (41.09G inference FLOPS per sample) for B-32 and B-16, respectively.

For the text-to-video experiments, Noting that estimating the FLOPS for some models in Table 3 and Table 12 poses several challenges due to the lack of publicly available information. For example, some of their training codes, and training datasets have not been open-sourced, and specific details such as the resolution and frame rate of training videos are unavailable. Thus, we report the estimated FLOPS for the models with sufficient details about the training samples and open-sourced models below. We utilized the inference code of their released open-source models and measured the FLOPS during inference using the calflops [3] tool. We

---

[3]https://github.com/MrYxJ/calculate-flops.pytorch

Table 12: Leading models on the VBench leaderboard as of Sep 23, 2024. "Board Avg." denotes the weighted average of normalized scores across 16 metrics defined by VBench. "Quality Avg." represents the aggregated scores from 7 video quality metrics, whereas "Semantic Avg" aggregates scores from 9 metrics evaluating the consistency between prompts and generated videos. "Uniform Avg." indicates the simple average of scores related to quality and semantic metrics. *Data-Juicer (T2V, 147k)* is based on T2V-Turbo distillation training on 147k instances. In order to demonstrate the scalability of our methodologies, we then further self-distilled our model to obtain *Data-Juicer (DJ, 228k)* on other 228k instances.

| Models (Ranked by leaderboard) | Board Avg. (%) | Uniform Avg. (%) | Quality Avg. (%) | Semantic Avg. (%) |
|---|---|---|---|---|
| 1. **Data-Juicer (DJ, 228k)** | **82.53** | **81.26** | 83.38 | **79.13** |
| **Data-Juicer (T2V, 147k)** | 82.10 | 80.54 | 83.14 | 77.93 |
| 2. Gen-3 (RunwayML, 2024) | 82.32 | 79.64 | **84.11** | 75.17 |
| 3. VEnhancer (VC2) (He et al., 2024a) | 81.97 | 80.00 | 83.27 | 76.73 |
| 4. Kling (2024-07) (Kuaishou, 2024) | 81.85 | 79.54 | 83.39 | 75.68 |
| 5. LaVie-2 (Wang et al., 2025) | 81.75 | 79.50 | 83.24 | 75.76 |
| 6. CogVideoX-5B (Yang et al., 2024) | 81.61 | 79.90 | 82.75 | 77.04 |
| 7. Vchitect 2.0-2B (Fan et al., 2025) | 81.57 | 80.15 | 82.51 | 77.79 |
| 8. T2V-Turbo (VC2) (Li et al., 2024a) | 81.01 | 78.67 | 82.57 | 74.76 |

Table 13: Detailed FLOPs comparison for representative T2V models. The factor $\alpha$ indicates a constant taking into account the disclosed quantity of training data.

| Models | Performance | Full size of used Dataset | Inference FLOPS per Sample | (Estimated) Number of Training Samples | (Estimated) Training Cost in EFLOPS |
|---|---|---|---|---|---|
| Ours (stems from VC2) | 82.53 | 228k | 12T/sample, 320p * 8fps | **640k** | $7.68\alpha$ |
| Ours (stems from VC2) | 82.10 | 147k | 12T/sample, 320p * 8fps | **640k** | $7.68\alpha$ |
| VEnhancer (stems from VC2) | 81.97 | 350k | 478T/sample, 720p * 24fps | $\geq$ **350k** | $\geq 167.3\alpha$ |
| CogVideoX-5B | 81.61 | 1,051M | 338T/sample, 720p * 8/16fps | **35M** | $\geq 11841\alpha$ |
| Vchitect 2.0-2B | 81.57 | - | 252T/sample, 768p * 8fps | - | - |

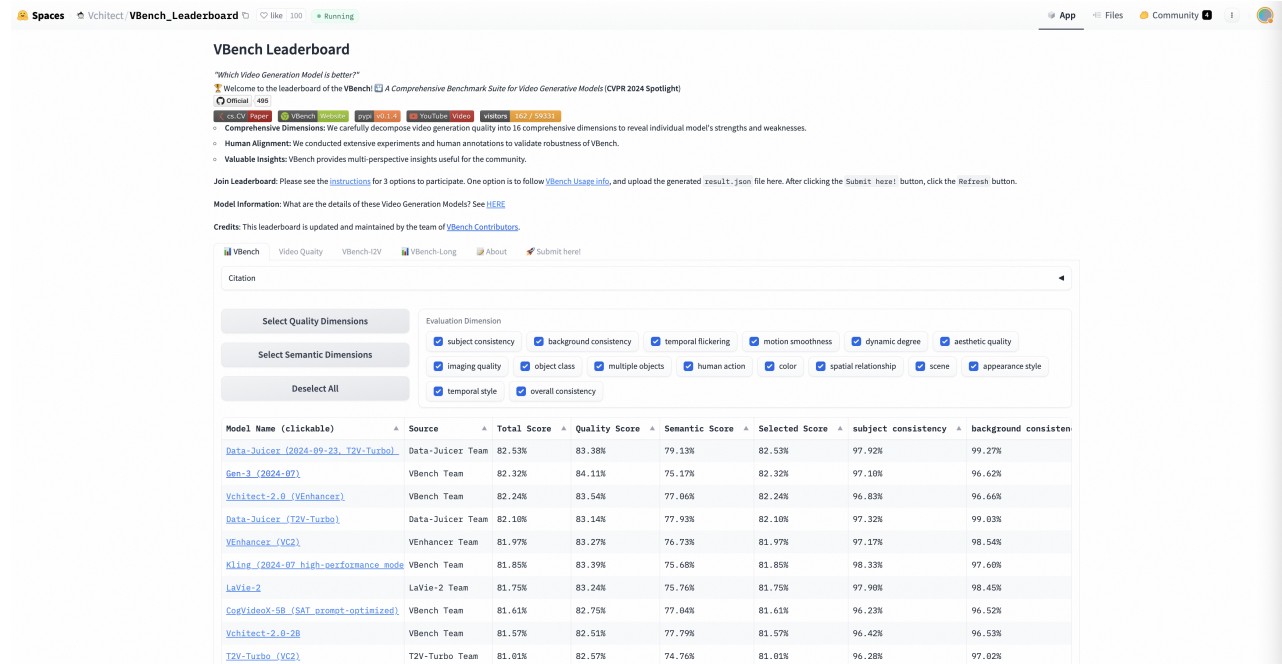

Figure 13: The VBench Leaderboard as of September 23, 2024, illustrating the rank 1 achievement empowered by our data-model co-development workflow.

then estimated the training FLOPS by applying a factor of $\alpha$ and taking into account the disclosed quantity of training data. From this, we can see that our method achieves strong performance while utilizing fewer FLOPS, especially in comparison to the third-place model, VEnhancer, which also originates from VC2, the same upstream model as ours. The second-place Gen-3 and the fourth-place Kling are omitted as they are proprietary models within commercial products.

### E.10. Iterative Workflows in Sandbox

Data-Juicer Sandbox allows iterative workflows, where data-model co-development is activated iteratively to enable data and model improvements for each other. We conduct a further iterative experiment based on the basic single-operator experiments for the image captioning fine-tuning task in Sec. 4.2, where the full experimental results are shown in Table 8.

**Experiment Settings.** We regard the single-operator recipes in the basic experiments as the $recipe_1$, and their trained models as $checkpoint_1$. Based on these first-iteration results, we select the best checkpoint from the best single operator `Text Length` to conduct a second iteration of single-operator continuous training for all single-operator recipes $recipe_2$ and obtain the second-iteration trained models $checkpoint_2$. Finally, we evaluate the $checkpoint_2$ to find out the impacts of the second-iteration data-model co-

development on the $checkpoint_1$.

**Results.** As the Table 14 shows, the second iteration continues to improve the model performance. Notably, while the ranking of OP effects changes slightly between iterations, most effective OPs remain consistent: 8 out of the top-10 OPs in the second iteration overlap with those from the first iteration; 7 of top-10 OPs in the first iteration increase their rankings by an average of 6.14 positions and 3 of them decrease by an average of 3.33 positions. Using the same OPs, data pool performance changes, providing actionable insights for dynamic environments.

This also highlights promising future directions and underscores the distinction between our work and DataComp. While DataComp primarily focuses on CLIP pretraining and scaling compute with duplicated data, our data recipes are validated across a broader range of tasks, including text-to-video generation (EasyAnimate, T2V-Turbo), image-to-text pretraining (MGM), post-training (InternVL-2.0) and the iterative co-development.

### E.11. Operators beyond Filters

There are diverse types of OPs in Data-Juicer in addition to Filters. In this section, we conduct several experiments on two representative Mappers using MGM-2B on the image-to-text task to verify the effectiveness of operators beyond Filters of the sandbox.

**Experiment Settings.** We use the same experiment setting of single-operator experiments for the image-to-text task in Sec. 4.2, except that we replace the single-operator recipes of Filters with two new recipes of two Mappers: `image_diffusion_mapper`: it regenerates a new image for each sample based on the given caption; `image_captioning_mapper`: it recaptions the existing image in each sample.

**Results.** As the Table 15 shows, images generated by `image_diffusion_mapper` significantly improve the model performance compared to the original images. In Fig. 14, it highlights a visual example before and after processing with these two Mappers, with the core objects in the captions marked in red. We can observe that diffusion models used in the `image_diffusion_mapper` effectively locate and better present the key objects (e.g., "setting sun") in the newly generated image, and it also removes redundant information (e.g., watermarks, link texts). We think this kind of "denoising" capability of `image_diffusion_mapper` contributes to performance gains. Experiments on other kinds of OPs are left to future work.

### E.12. Potential Model Development: Automated Prompt Adjustments

Most parts of this paper focus on the data improvements in data-model co-development. In this section, we present several experiments for automated prompt adjustment to show the potential model development part of the Data-Juicer Sandbox.

**Experiment Settings.** We select the image captioning fine-tuning task, extend a new model inference hook with about 40 lines of code changes, and update its prompt configuration file. Based on the iterative experiment results in Sec. E.10, we use the top-1 recipes found in two iterations. Then we select 10 different prompts generated by Qwen2.5-max and study the effects of them on trained models in different iterations.

**Results.** As the Table 16 summarizes, the ranking and performance of these prompts vary significantly across iterations, ranging from -1.673 to +0.944. Some of them beat the baseline prompt, and some of them can enhance or degrade performance in both iterations. It seems that the most suitable prompt is changing during the iterative data-model co-development, which indicates the potential for auto-optimizing model configurations in the context of the co-development of the Data-Juicer Sandbox.

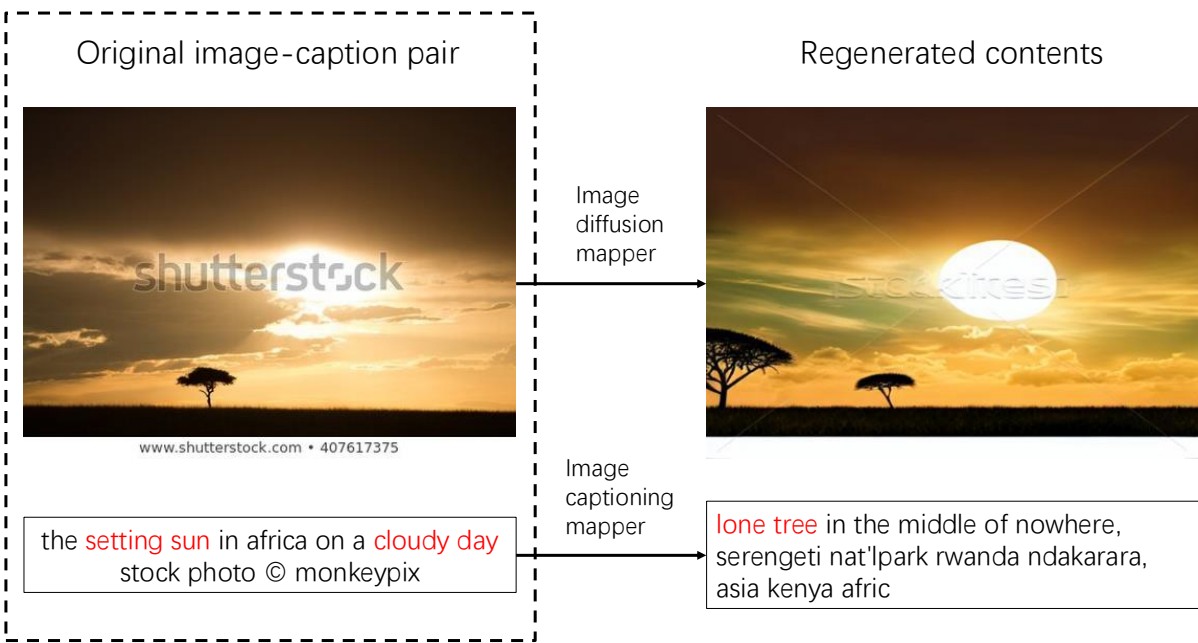

Figure 14: An example of image and caption pair before and after being transformed by `image_diffusion_mapper` and `image_captioning_mapper`.

Table 14: Top-10 OP performance ranking with 2 iterations, including their statistical dimensions and the improvements relative to the baselines for the image captioning task. The 2nd iteration is continuously trained on the top-1 OP (*Text Length*) checkpoint in the 1st iteration and the improvements are relative to it as well. The OPs in bold in the 2nd iteration also appear in top-10 OPs of the 1st iteration.

| 1st iter. | | 2nd iter. | |
| --- | --- | --- | --- |
| OP-Generated Statistics | Average Performance Changes (%) | OP-Generated Statistics | Average Performance Changes (%) |
| *Text Length* | 0.760 | *Flagged Word Ratio* | 0.607 |
| *Image Watermark Score* | 0.477 | ***CLIP Image-Text Similarity*** | 0.599 |
| *Character Repetition Ratio* | 0.449 | ***Text Perplexity*** | 0.521 |
| *Text Perplexity* | 0.331 | ***Character Repetition Ratio*** | 0.434 |
| *Word Repetition Ratio* | 0.300 | ***Image Aesthetics Score*** | 0.420 |
| *Image Width* | 0.265 | ***Image Size*** | 0.376 |
| *CLIP Image-Text Similarity* | 0.225 | ***Phrase Grounding Recall*** | 0.358 |
| *Phrase Grounding Recall* | 0.188 | ***Image Width*** | 0.320 |
| *Image Aesthetics Score* | 0.186 | *Image Height* | 0.285 |
| *Image Aspect Ratio* | 0.171 | ***Image Watermark Score*** | 0.264 |

Table 15: The performance improvements of two mapper OP, *image_diffusion_mapper* and *image_captioning_mapper*, relative to the baseline, which regenerates a new image/caption for each sample using diffusion/BLIP models respectively. The baseline is based on the original raw images.

| Mapper OP | Average Performance Changes (%) |
|---|---|
| *image_diffusion_mapper* | **13.45** ± 11.33 |
| *image_captioning_mapper* | -1.07 ± 0.65 |

Table 16: The experimental results of automated prompt adjustment on the top-1 recipes found in 2 iterations with 10 different prompts for the image captioning task. The baseline prompt is the default prompt of InternVL2.5. The values that are in **bold** or underlined are the best or the 2nd best ones in each column.

| Prompt | Average Performance Changes (%) | | 1st vs. 2nd |
|---|---|---|---|
| | 1st iter. | 2nd iter. | |
| (*Baseline*) Provide a one-sentence caption for the provided image. | 0.760 | **0.607** | > |
| Provide a brief yet comprehensive caption that highlights the primary elements and overall mood of the image. | **0.944** | -0.705 | > |
| Describe the image in one sentence, focusing on the main subject and its surrounding environment. | 0.832 | -0.241 | > |
| Summarize the scene depicted in the image with a concise and descriptive sentence. | 0.734 | 0.148 | > |
| Summarize the content of the image in one clear and accurate sentence without omitting important details. | 0.542 | 0.186 | > |
| Describe the most important details of the image in one sentence. | 0.400 | 0.130 | > |
| Craft a concise image caption highlighting the main activity alongside coexisting static elements in the image. | 0.083 | -1.622 | > |
| What is the main subject of the image, and how would you describe it in one sentence? | -0.026 | -0.463 | > |
| If there is movement or activity in the image, summarize it in a single descriptive sentence. | -0.594 | 0.433 | < |
| Please provide a one-sentence caption that describes the primary subject's action and its relationship with the surrounding environment. | -1.578 | -1.510 | < |
| What stands out most in the image? Capture it in a concise and striking one-sentence caption. | -1.673 | -1.300 | < |

