# OpenReview forum: "Data-Juicer Sandbox: A Feedback-Driven Suite for Multimodal Data-Model Co-development"
_ICML.cc/2025/Conference — ICML 2025 spotlightposter_

### Official Review · Reviewer_5ZDF · 2025-03-13

**Overall Recommendation:** 4

**Summary:**

The paper introduces Data-Juicer Sandbox, an open-source suite that supports co-development of multimodal data and models. It proposes a feedback-driven approach in which data processing and model training are iterated together, rather than in isolation.

A central component is the “Probe-Analyze-Refine” workflow, where data are filtered or enriched using various operators, then tested with smaller reference models. Feedback is collected on both data characteristics (such as quality and diversity) and model metrics (across multiple benchmarks), leading to insights on how specific data operations affect performance. These insights guide incremental recipe-building steps, combining or scaling up promising data operations to improve model outcomes at a lower computational cost.

Through experiments on three multimodal tasks (I2T generation, T2V generation, and I-T pre-training), the paper reports consistent gains in performance (including a top entry on the VBench leaderboard). The work further highlights how these data-driven refinements can transfer to larger-scale models and training runs, showing data efficiency gains and reinforcing the synergy between well-designed data curation and model training.

**Claims And Evidence:**

The paper’s core claims—that a unified sandbox improves multimodal data and model co-development by (1) revealing useful data operations via small-scale experiments, (2) reducing trial-and-error costs through incremental “probe-analyze-refine,” and (3) boosting performance on tasks such as image-to-text, text-to-video, and image-text pre-training—are mostly backed by clear experimental outcomes. For example, the text-to-video experiments include leaderboard results on VBench, which supports the performance gains they claim. The authors also provide cost comparisons and scaling analyses, lending some credibility to the claim of more efficient and systematic co-development.

Potential gaps lie in: 1) Generality: Most evidence comes from three tasks; more diverse tasks or larger sets of models could further strengthen or limit the conclusions. 2) Comparative baselines: Although they compare with existing methods (e.g., Runway’s Gen-3 or other pretraining strategies), it is not always clear that the baseline setups or alternative data-processing methods were exhaustively explored.

Overall, while the main claims are supported by strong empirical findings on chosen tasks, further evidence across broader data types or tasks could make the presented framework’s benefits even more convincing.

**Essential References Not Discussed:**

The paper does not cite Ling et al. (2025), "Diversity as a Reward" (DAAR), which introduces a structured framework for optimizing data diversity during fine-tuning. DAAR provides an automated approach to balance inter-domain and intra-domain diversity, using model-driven feedback to guide data selection. This contrasts with Data-Juicer's human-guided, iterative approach to dataset refinement.

Additionally, JEST (Evans et al., NeurIPS 2024) formulates data selection as a dynamic optimization problem, significantly reducing training iterations by prioritizing high-value samples. This aligns with Data-Juicer’s goal of efficient data-model co-development but takes a different approach by embedding selection directly into the training loop.

Ling, Zhenqing, Daoyuan Chen, Liuyi Yao, Yaliang Li, and Ying Shen. "Diversity as a Reward: Fine-Tuning LLMs on a Mixture of Domain-Undetermined Data." arXiv preprint arXiv:2502.04380 (2025).
Evans, Talfan, Nikhil Parthasarathy, Hamza Merzic, and Olivier Henaff. "Data curation via joint example selection further accelerates multimodal learning." Advances in Neural Information Processing Systems 37 (2024): 141240-141260.

**Experimental Designs Or Analyses:**

The paper’s “probe–analyze–refine” workflow is generally well-conceived: they start with single-operator data pools to observe isolated effects, combine top-performing operators, and then scale data. This incremental approach reduces trial-and-error costs while producing comparative feedback. They also use a consistent set of baseline models and hyperparameters so that only the effect of data changes is measured.

One caution is that they rely on single-pass experiments (with no mention of repeated runs) to quantify the impact of each operator or recipe. While this is typical for many large-scale training studies (due to resource limits), it means we don’t see variance estimates that might highlight sensitivity to random seeds or sampling. Additionally, there is an assumption that small-scale subset behavior extrapolates to larger scales. They do provide a brief theoretical justification and some empirical scaling analyses, but distribution shifts or non-i.i.d. sampling could reduce perfect transfer.

In short, the design is sensible and largely valid for controlled comparisons, but it would be stronger with additional runs or variance reporting to confirm each operator’s reliability in improving results.

**Methods And Evaluation Criteria:**

The paper’s proposed probe–analyze–refine workflow aligns well with the intended goal of iteratively improving data and model quality. The authors use task-relevant metrics (e.g., MMBench for image-to-text, VBench for text-to-video, and standard contrastive-learning metrics for image-text pretraining), which are fairly standard and reflect the models’ target abilities. These evaluation criteria effectively show how data interventions translate into performance gains.

A minor limitation is that most experiments are within three canonical tasks, so the broad generalizability of the chosen metrics or methods could be explored further. Even so, the approach is well-reasoned and uses suitable benchmarks to highlight measurable improvements.

**Other Comments Or Suggestions:**

1. While the blueprint for combining data-centric methods (e.g., filtering, annotation generation) with model-centric improvements is very appealing, the Data-Juicer Sandbox as presented does not clearly include many mechanisms for the model’s evolving needs. In other words, there could be additional iterative routines where model improvements inform new data requests or automated prompt adjustments. Explaining how the pipeline might be extended to facilitate continuous model-data evolution would strengthen the paper’s vision.
2. Consider adding a “best practices” subsection in the appendix, summarizing lessons about which operators and metrics are especially critical under different conditions. This might serve as a quick reference for new sandbox users.
3. The text occasionally references “industry-level” or “production-ready” usage; clarifying any special hardware or cluster requirements would make it clearer how to adopt this in large-scale settings.
4. Minor editorial note: Some references in the bibliography appear slightly inconsistent in formatting (e.g., capitalization in titles). It may help to standardize them.

**Other Strengths And Weaknesses:**

Strengths:

1. The paper proposes a tangible, open-source sandbox framework that addresses data-model co-development – a recognized but underexplored area in multimodal research.
2. The “Probe–Analyze–Refine” strategy is clear, methodical, and well-presented. It extends beyond high-level theory by offering a practical pipeline (with code) that other researchers can use.
3. Empirical results underscore the potential cost savings of systematically iterating on small data subsets before scaling up – valuable in a field where full-model training costs can be huge.
4. Demonstrations in image-to-text, text-to-video, and image-text pretraining are reasonably diverse, strengthening the paper’s impact and showing general applicability.

Weaknesses:

1. No Formal Validation of Combining Operators:
While the paper clearly shows individual operators’ effects, the process of stacking multiple top operators sometimes degrades performance. It remains unclear why certain combinations interact negatively. For instance, a multi-operator strategy could preserve only a narrow slice of data distribution or inadvertently filter out complementary samples. A deeper exploration of these “operator interaction” phenomena (e.g., synergy vs. conflict) would strengthen the paper’s argument that combining top operators is not trivially beneficial.
2. Limited Empirical Repetitions:
The paper often uses single-run experiments for each operator/recipe, presumably to control computing. However, large-scale training can show variance in results (random seeds, sampling). Without repeated trials or uncertainty estimates, it is difficult to assess how reproducible (or fragile) these improvements are. The text occasionally notes that cost constraints make multiple runs challenging, but even small-scale tests repeated 2–3 times would add confidence that results do not hinge on luck.
3. Unclear Impact of Distribution Shifts:
The framework assumes that small-pool “probe” experiments reliably anticipate behavior at full scale. Yet in practice, distribution shifts or domain mismatches could exist between the reduced sampling and the real-world training set. In particular, data rebalancing or filtering might produce biases that only manifest when the entire large dataset is used. More discussion or case studies on how best to mitigate such shifts would be valuable.

**Questions For Authors:**

1. Why All Operators Are Filters? The paper positions several “operators” (OPs) as different filtering or perplexity-based selection procedures. It is not fully clear whether the pipeline includes other augmentation methods or more diverse operators (e.g., paraphrasing Q&As, image transformations). Highlighting (or adding) such diversity might boost the effectiveness and generality of this work.
2. Could you elaborate on why the highest-performing individual operators sometimes fail to improve results when combined? Are certain filters redundant, or do some systematically remove data that others rely on? How might future versions of the sandbox help diagnose when operators conflict?
3. Do you have preliminary variance estimates (e.g., dev set performance standard deviations) for a small subset of your probe experiments? If so, how large are the fluctuations? Knowing approximate variability would help judge whether a single-run improvement of, say, +1% is robust or possibly random.
4. Have you found any real-world cases where an operator looks beneficial in small-pool settings but fails to extrapolate at full scale (or vice versa)? If so, how do you suggest researchers detect and mitigate such mismatches early on?
5. One appealing application would be continuously refining data as new samples appear (e.g., updated data streams). Would your sandbox accommodate dynamic additions to the dataset, or is it currently designed only for static snapshots?
6. You mention “industry-level” usage in passing. Do you envision a canonical workflow for large organizations to adopt Data-Juicer Sandbox, or do you foresee each group creating highly customized pipelines? A brief discussion of how you see this platform evolving in real deployments would be illuminating.
7. Because reproducibility and generalization are important for future work, could you share more implementation details or guidelines to ensure that this framework can be adapted to different tasks? How to ensure the complex codebase can be easily adapted and reused.
8. I intend to give a rating of "accept" due to the potential impact of this work. However, I am still concerned about the reusability of this work. I will consider raising my rating further if all concerns are addressed properly.

**Relation To Broader Scientific Literature:**

### Multimodal Model Training: Traditional vs. Co-Development

- **Traditional Training:** Treats data and model design separately; data is fixed before training, limiting feedback.
- **Co-Development:** Data-Juicer integrates data and model training through an iterative “Probe-Analyze-Refine” workflow, improving both simultaneously.

---

### Data-Centric AI vs. Data-Juicer

- **Traditional Data-Centric AI:** Focuses on dataset curation and filtering before training, lacking real-time feedback.
- **Data-Juicer:** Iteratively refines datasets using model performance, optimizing data selection dynamically.

---

### Cost-Efficient Co-Development

- **Challenges:** Large models require costly computation; efficient iteration is crucial.
- **Data-Juicer’s Solution:** Uses small-scale experiments before full training, reducing unnecessary compute.

---

### Optimizing Data Selection

- **Data-Juicer’s Insight:** Task-specific data strategies—image-text models benefit from diversity, video models need high-quality samples.
- **Compared to Other Methods:** Unlike RL and gradient-based selection, Data-Juicer enables empirical testing of data modifications.

Both reinforce feedback-driven data selection, optimizing dataset quality alongside model training.

**Theoretical Claims:**

The authors do provide a concise statistical bound (inAppendix C) that links performance changes observed in small-pool experiments to those expected at full scale, using a standard concentration inequality. Their derivation assumes (1) a simple i.i.d. sampling scheme from the original dataset and (2) that the relevant performance changes can be bounded within a fixed interval [a, b]. Under these assumptions, their stated probability bound,

$
P[\Delta_{\text{pool}} - \mathbb{E}[\Delta_{\text{full}}]\ge\epsilon] \le
\exp\bigl(\tfrac{-2\epsilon^2}{(b-a)^2}\bigr),
$

is a straightforward application of Hoeffding’s inequality. The argument appears correct on its face, though it is relatively high-level and does not deeply address issues like data distribution shifts or non-i.i.d. data. In other words:

- If the dataset and performance metrics meet the i.i.d. assumptions, the bound is formally correct.
- If there is strong distribution mismatch or outlier-heavy data, the theoretical guarantee would require additional considerations.

The paper does not provide deeper or more elaborate proof, and no immediate errors stand out. The result is intended mainly as a heuristic justification for why small-pool insights might translate to the full dataset.

---

> ### Author Rebuttal · Authors · 2025-04-01
>
> We sincerely appreciate your time, thorough evaluation and valuable feedback! Below, we address all your raised Weaknesses (W), Comments (C), Questions (Q) and Suggestions (S).
>
> > New results: https://anonymous.4open.science/r/icml_submission8414-7C89/rebuttal.pdf
>
> ---
>
> ## [C on Claims & Methods] "Potential Gaps in Generality ... are within three tasks"
> We have added new experiments to enhance generalization verification, including scaling on the InternVL series (1B, 2B, 4B, 26B) and a new captioning task. For details, see reply to Reviewer CLJN.
>
> ---
>
> ## [C on Variance, W2 & Q3] "... Without repeated trials ... "
>
> You may have missed this detail (e.g., error bars in Fig. 3). We have ran experiments 2–5 times, but included std in the appendix (lines 901, 991, Table 6) due to space limits. Note the reported performance reflects changes relative to *random pools*. Consistent *positive values* across trials demonstrate reproducibility and generalizability.
>
>
> ---
>
> ## [C on Literature; S1 & Q5] DaaR & JEST; "Iterative routines..."
> Thank you for pointing out these works! DaaR was released (Feb 5) after the ICML submission ddl (Jan 9) and already cites and builds upon our work (see its Sec B.4). JEST's dynamics and your mentioned iterative sandbox align with future directions we aim to explore. We will cite both works in the final version and expand discussions accordingly.
>
> Besides, during the rebuttal, we added new experiments supporting iterative training: starting with ckpt0 → identifying recipe1 → training ckpt1 → refining recipe2 → training ckpt2. Interestingly, while recipes evolved, ckpt2 still improved performance. For details, see reply to Reviewer 3zp6.
>
> ---
>
> ## [W1 & Q2] OP Combination: Synergy vs. Conflict, Failure vs. Detection
> Great suggestion! In Appendix E.3–E.5 (Table 8, Figs. 5–8), we provide quantitative analyses using Pearson correlations and entropy. We plan to extend this suite with automated and visualized lineage analysis tools. For example, t-tests can detect significant changes in filter outputs' stats dimensions after adjacent OPs. A practical use case: if an image-text matching filter notably reduces text length stats, it may indicate noisy captions affecting accuracy.
>
> ---
>
> ## [W3 & Q4] Impact of Distribution Shifts
> Thank you for this important comment! While addressing distribution shifts robustly requires substantial effort (left for future work), we highlight:
>
> 1. Empirically, our results already cover challenging distribution shifts (Table 5), where small-scale recipes benefit larger scales across model architectures, data, & compute.
>
> 3. Methodologically, the framework’s open-source extensibility allows flexible customization of workflows while reusing data/model infra (see Sec. B.1–B.3). New hooks can integrate advanced strategies like dataset distillation [1] and intermediate tensor shaping [2]. Tools mentioned in [W1, Q2] also offer intuitive signals.
>
> [1] Large scale dataset distillation with domain shift, ICML'24
>
> [2] Scaling for Training Time and Post-hoc Out-of-distribution Detection Enhancement, ICLR'24
>
> ---
>
> ## [S2~S4] Writing Improvements
> Thank you! We have provided hardware details (lines 927–992, 1037–1041) and will expand the "best practices" subsection with summarized OP-specific labels from Tables 6–7 and Figures 2, 9. We’ll also unify reference formatting as suggested.
>
> ---
>
> ## [Q1] OPs Beyond Filters (e.g., Paraphrasing Q&As, Image Transformations)
> Insightful question! Per your request, we conducted new experiments on two representative Mappers using MGM:
> - image_diffusion_mapper: Regenerates a new image for each sample based on the caption.
> - image_captioning_mapper: Recaptions the existing image in each sample.
>
> `Results`
>  - As shown in *Table 3* and *Fig. 3* of the new PDF, *image_diffusion_mapper* significantly improves performance compared to the original images.
>  - Fig. 3 highlights a visual example before and after processing with these mappers. Core objects in captions are marked in red. We observe: (1) Diffusion models effectively locate and better present key objects (e.g., "setting sun") in generated images. (2) They also remove redundant information (e.g., watermarks, link text), likely contributing to performance gains.
>
> ---
>
> ## [Q6 & Q7] Deployment & Sys Details
> - Appendix B discusses the architecture, capability factory, behavior hooks, extensibility. This decoupling enables rapid adaptation to new tasks while reusing workflows, helping us do quick new exps during rebuttal. The work is also already deployed in several industrial scenarios (specific entities undisclosed due to anonymity).
> - To simplify complexity, we are service-ifying it: RESTful APIs & MCP servers for OPs, unified & transparent environment switch for third-party modules. These efforts reduce programming complexity.
>
> ---
>
> ## [Q8] Reusability & Rating
> We will incorporate these improvements into final version and hope they strengthen your confidence. Thanks again!

---

> > ### Comment · Reviewer_5ZDF · 2025-04-03
> >
> > Thanks for the author's responses. Most of my concerns have been addressed. However, one of my personal questions has been ignored, i.g., "While the blueprint for combining data-centric methods with model-centric improvements is very appealing, the Data-Juicer Sandbox as presented does not clearly include many mechanisms for the model’s evolving needs. In other words, there could be additional iterative routines where model improvements inform new data requests or automated prompt adjustments. Explaining how the pipeline might be extended to facilitate continuous model-data evolution would strengthen the paper’s vision."
> >
> > Specifically, authors claimed that they present a new sandbox suite tailored for integrated model-centric and data-centric developments. So where is the model development part? I haven't seen any implementation in Data-Juicer Sandbox proposed for the convenience of model development yet. If my understanding is biased, please point it out also.

---

> > > ### Author Response · Authors · 2025-04-09
> > >
> > > > Thanks for the author's responses. Most of my concerns have been addressed. However, one of my personal questions has been ignored ...
> > >
> > > We appreciate your reply! Due to space limit, we consolidated answers under "[C on Literature; S1 & Q5]," which may have been too brief. Below, we address your question in more detail.
> > >
> > > ---
> > >
> > > ## Clarification on Model-Dev
> > >
> > > > So where is the model development part? I haven't seen any implementation in Data-Juicer Sandbox proposed for the convenience of model development yet. If my understanding is biased, please point it out also.
> > >
> > > We clarify that foundation model development involves three core phases: model selection (architecture), training (pre/post-training), and evaluation (foundational/downstream tasks). To streamline these stages, our sandbox acts as middleware, decouples and reuses existing infra from open-source community into all-in-one solutions.
> > >
> > > ### Design Perspective
> > >
> > > As detailed in Appendix B, we employ a modular architecture for these phases:
> > >
> > > - *Bottom Layer*: Factory classes integrate diverse training and evaluation libraries.
> > > - *Middle Layer*: Encapsulates model-agnostic behaviors like probe, train & evaluation hooks.
> > > - *Top Layer*: Implements workflows via ordered job lists, enabling users to adjust configuration files to define scenarios, hooks, metrics, and workflows.
> > >
> > > This design reduces the need to learn disparate libraries or start from scratch,
> > > allowing flexible reuse and combination of representative data-model dev modules:
> > >
> > > - For example, our experiments cover rapid prototyping across 5 architectures, ~80 metrics, and diverse pre-/post-training setups.
> > >
> > > - Thanks to its middleware design, the sandbox evolves with its underlying libraries, amplifying its utility through open-source contributions. For instance, with ModelScope-Swift & OpenCompass, it indirectly supports 500+ LLMs & 100+ eval datasets.
> > >
> > > The proposed "probe-analyze-refine" workflow is just one instantiation of sandbox's capabilities. More workflows can be extended while reusing pre-built codes, as illustrated later.
> > >
> > > ### Code-Level Efforts
> > > To simplify user actions and reduce programming complexity, we provide
> > > - One-line scripts to run the sandbox and adjust cfgs.
> > > - Dynamic dependency resolution and containerization for auto-setup.
> > > - Lazy-loader of Python packages for easy module loading.
> > > - Unified scripts to handle environment preparation, with third-party modules organized as services, each with its own entry program, and mapping to conda cfgs.
> > >
> > > ---
> > >
> > > ## Extending the Pipeline
> > >
> > > > There could be additional iterative routines where model improvements inform new data requests or automated prompt adjustments. Explaining how the pipeline might be extended to facilitate continuous model-data evolution would strengthen the paper’s vision.
> > >
> > > Beyond the flexibility of layered architecture, we demonstrate extensibility with two examples aligned with your suggestion:
> > >
> > > ### Case 1. Model Improvements Inform New Data Requests
> > > - `How to Extend` By wrapping a configurable loop around the top-level workflow (5–10 lines of code changes), we reuse the "probe-analyze-refine" workflow to enable iterative model improvements.
> > > - `New Example Added During Rebuttal`
> > >   - Setting: Starting with *ckpt0*, we refined *recipe1* → trained *ckpt1* → refined *recipe2* → trained *ckpt2*. The 1st iter aligns with single-OP exp from previous tasks. We then selected the best ckpt and applied top OP to original dataset as baseline.
> > >   - Results (summarized in Table 2 of uploaded PDF):
> > >     - The 2nd round further improves performance.
> > >     - Compared to the 1st round:
> > >       - *2 new OPs* emerge in the top-10.
> > >       - *7/10 OPs* increase their rankings by an average of *6.14 positions*.
> > >       - *3/10 OPs* decrease their rankings by an average of *3.33 positions*.
> > >     - Using the same OPs, the perf of data pools changed, providing actionable insights into data processing needs for dynamic environments.
> > >
> > > ### Case 2. Model Improvements Inform Automated Prompt Adjustments
> > > - `How to Extend` By adding a model inference hook in the middle layer (~40 lines of code changes) and updating its prompt cfg file, we can reuse the infra to study and optimize prompts.
> > > - `New Example Added During Rebuttal`
> > >   - Setting: Using top-1 recipes found in these two iters, we studied the effects of 10 different prompts, which were auto-generated by Qwen2.5-max with "You are a prompt expert, plz optimize the given one: {pmt}."
> > >   - Results (summarized in Table 4 of uploaded PDF):
> > >     - The ranking and performance of these prompts vary significantly across iterations (e.g., ranging from -1.673 to +0.944).
> > >     - Some prompts beat the baseline.
> > >     - Some prompts enhance/degrade performance in both iters, while others show inconsistent behavior: performing better in the 1st round (>0) but worse in the 2nd round (<0), or vice versa.
> > >   - These findings highlight sandbox's potential for auto-optimizing model configurations in the context of co-development

---

### Official Review · Reviewer_Dvoe · 2025-03-14

**Overall Recommendation:** 2

**Summary:**

This paper introduces the data juicer sandbox, an open-source suite that analyzes various metrics and make use of heuristics to facilitate the integrated development of multimodal data and models. The proposed "Probe-Analyze-Refine" workflow was validated through image-text pre-training with CLIP, MLLMs, and text-to-video generation. The `feedback-driven` approach, manually executed rather than automated, yields performance improvements and provides insights into the interplay between data quality, diversity, model behavior, and computational costs.

**Claims And Evidence:**

The primary contribution of the paper lies in its observations derived from comprehensive small-scale exploration. However, the generalizability of these conclusions, derived in  low data regimes, is questionable, particularly during specific model and task stages (eg the pretraining of a 2B MLLM). It remains uncertain whether these insights can effectively scale up with increased model parameters and data volume.

Furthermore, the conclusions are not expressed with precision or sufficient validation . For example, the definitions of quality and diversity are based on the performance/feedback of minitest on the target tasks, with the number of combinations of different quality levels serving as the diversity metric, rather than widely acknowledged measures such as embedding or domain specification. The definition of diversity, based on varying OP scores or the entropy used in Sec.E3, works like simple bin sampling in traditional AL, may work but not effective, which makes the claim less persuasive.
Besides, the conclusion differ from previous findings, such as Scaling Laws for Data Filtering -- Data Curation cannot be Compute Agnostic, CVPR24.

**Essential References Not Discussed:**

Scaling Laws for Data Filtering -- Data Curation cannot be Compute Agnostic, CVPR24.
Active learning for convolutional neural networks: A core-set approach. ICLR 2018.

**Experimental Designs Or Analyses:**

Yes I've checked the validity of the proposed experimental designs. The experiments conducted are comprehensive. My concerns are listed above.

**Methods And Evaluation Criteria:**

The cost of using the sandbox to identify high-quality training data is described as (1 + mr) ≤ M, where M represents the epochs required for training on the original data, m is the number of epochs times the number of OPs on the subset, and r is the subset ratio. The authors claim this cost is "feasible," yet it appears impractical. In fact, the cost of searching for optimal hyper-OPs might exceed that of simply training on the full dataset. This issue is common in data selection work, such as
Diddee etal. Chasing Random: Instruction Selection Strategies Fail to Generalize, 2024
Xiaetal. Rethinking Data Selection at Scale: Random Selection is Almost All You Need, 2024.

**Other Comments Or Suggestions:**

no further comments

**Other Strengths And Weaknesses:**

The authors' engineering implementation and comprehensive experiments covering three different tasks should be acknowledged.

**Questions For Authors:**

no further questions

**Relation To Broader Scientific Literature:**

The significance of the task and the motivation behind the work are both significant. However, there are doubts regarding the practical applicability of the experimental design and the generalizability of the conclusions drawn from the study, as discussed above.

**Theoretical Claims:**

There is no theoretical claims in the manuscript.

---

> ### Author Rebuttal · Authors · 2025-04-01
>
> We sincerely appreciate your time, insightful feedback, and recognition of the work's significance! Below, we address your comments point by point.
>
> > The new results: https://anonymous.4open.science/r/icml_submission8414-7C89/rebuttal.pdf
>
> ### [Para 1 in Claims part] "generalizability is questionable, ..., particularly during specific model and task stages (eg the pretraining of a 2B MLLM)"
> Per your suggestion, we conducted new experiments on a larger model suite (*1B, 2B, 4B, 26B InternVL*) and a different task (*fine-tuning for captioning*). For details, plz see our response to Reviewer cLJN.
>
> Collectively, extensive experiments demonstrate the generalizability, covering different
> - *Scenarios*: Text-to-video generation, image-text pre-training, image-to-text generation for general image understanding & image captioning, iterative enhancement (see response to Reviewer 3zp6).
> - *Stages*: both pre-training and fine-tuning.
> - *Model Architectures*: EasyAnimate, T2V-Turbo, MGM, InternVL2, and CLIP.
> - *Scales*: Model parameters, data sizes, and compute FLOPs.
> - *Data Processing Solutions*: over 40 Data-Juicer Filters and Mappers (see response to Reviewer 5ZDF)
> - *Model Feedback*: 78 metrics.
>
> ---
>
> ### [Para 2 In Claims part] "... measures such as embedding or domain specification ... " "differ from previous findings ... CVPR24"
>
> `More acknowledged measures`
>
> Following your suggestion, we conduct new experiments to dive into the training data in the text-to-video task with a tagging model, extracting core objects, concepts, and domains. We analyzed the domain-specific distribution of selected examples (*video_nsfw_filter*) via word clouds of these tags. Results are in Fig. 2 of the uploaded PDF.
>
> As we can see, although the data pool with high video NSFW scores is the most diverse one, our model achieves the best performance on the data pool with low video NSFW scores, which is consistent with the argument presented by the left column of lines 296-298 in our submission.
>
> `The CVPR24 work`
>
> Thanks to bring this work into our attention. We respectfully highlight that the differences in conclusions stem from distinct experimental settings, which we believe lead to complementary insights:
>   1. **Data Curation Setup:**
>    In their most relevant scaling experiments (Figs 5 & 11), the authors examine data filtering strategies by varying the *"quantile thresholds"* while keeping the filtering operator fixed. In contrast, we explore *different filtering operators* while maintaining the same "quantile threshold."
>   2. **Task and Compute Setup:**
>    Their work mainly focuses on DataComp and CLIP pretraining task, scaling compute with duplicated data. Beyond CLIP, we validate our data recipes across more tasks, including text-to-video generation, image-to-text pretraining, and post-training. For a detailed comparison, please refer to Fig.3 & Table 5 in our original paper, and the newly added InternVL experiments mentioned above.
>
> ---
>
> ### [Methods part] "feasible, ... yet it appears impractical." "This issue is common in data selection work, such as ..."
>
> `HPO considerations`
>
> We acknowledge that the cost analysis in our claim omits the coefficient introduced by HPO, under the practical assumption that *HPO is equally required for fair comparisons in large-scale scenarios*. In fact:
> - Large-scale systems often introduce additional complexities (e.g., debugging in distributed clusters, hardware/software errors), which can inflate the HPO coefficient significantly.
> - Industry practices typically rely on small-scale experiments to predict or generalize HPO for larger settings, as discussed in [1].
>
> [1] *Predictable Scale: Part I — Optimal Hyperparameter Scaling Law in Large Language Model Pretraining.*
>
> `The work "Random Selection is Almost All You Need"`
>
> We thank the reviewer for pointing out it. However, our work differs from it notably in both scope and baseline comparisons:
> 1. **Different Scenarios:**
>    - Our work focuses on *four diverse multimodal tasks* (text, image, video), whereas the cited works primarily study *pure text post-training*.
>    - Our data selection strategy is derived from *40+ operators and 70+ performance metrics*, offering a broader and more adaptive framework compared to specific strategies studied in this work, like GPT-based scoring or heuristic methods (e.g., longest instruction).
>
> 2. **Empirical Evidence of Feasibility:**
>    - All numerical results in our small-scale experiments are benchmarked against a *Random baseline pool of equivalent data size* (lines 216 & 227).
>    - The positive values reported in Table 3 and the Flops comparison across different scales demonstrate the effectiveness of our method in identifying solutions that consistently outperform random selection across various scenarios.
>
> ---
>
> We appreciate the  valuable feedback and hope that these responses can address your comments, leading you consider an increase in the rating. Thank you once again!

---

### Official Review · Reviewer_s5fL · 2025-03-21

**Overall Recommendation:** 3

**Summary:**

This paper introduces a sandbox suite with a feedback-driven experimental platform, which supports cost-effective iteration and guided refinement of both data and models. The authors conduct experiment on image-to-text generation, text-to-video generation and image-text pretrianing. The results demonstrate the proposed sandbox can effectively improve dataset and models, and also give valuable insights into the relationship data processing and model performance.

**Claims And Evidence:**

The paper claims that the proposed sandbox effectively achieves data-model co-development. However, only the contribution to data refinement has been proved, while how the sandbox improves model development compared to other platforms remains unclear.

**Essential References Not Discussed:**

No.

**Experimental Designs Or Analyses:**

The evaluation mainly focuses on the sandbox's application in data refinement, which may not be enough to demonstrate its superiority over other platforms.

**Methods And Evaluation Criteria:**

The proposed method for data and model development is reasonable. However, the evaluation mainly focuses on the sandbox's application in data refinement, which may not be enough to demonstrate its superiority over other platforms.

**Other Comments Or Suggestions:**

It would be better to improve the organization of the paper to enhance readability.

**Other Strengths And Weaknesses:**

Strengths:
   1. The paper proposes a platform for efficient data and model development, which is valuable for improving research efficiency.
   2. The paper analyzes the impact of different data recipes on model performance and provides several valuable insights.

Weaknesses:
   1. Although the paper claims the proposed sandbox effectively enables data-model co-development, only the contribution to data refinement has been proved, how the sandbox improves model development compared to other platforms remains unclear.
   2. Data recipes obtained from small-scale experiments may not generalize to large-scale experiment, thus its challenging to determine the effectiveness of the sandbox on large-scale training.
   3. The organization of the paper is somewhat confusing. I think the sandbox's properties should be the core focus, but there are many paragraphes talking about observations from data experiment. This makes it hard for readers to focus on the key advantages of the sandbox.

**Questions For Authors:**

1. I am curious to learn about how the data recipes obtained from this sandbox compare to the optimal data recipes in large-scale experiments.
2. What are the advantages of the sandbox in model refinement?

**Relation To Broader Scientific Literature:**

The authors introduce a new sandbox for cost-effective data and model refinement, which can enhance the efficiency of model development and is valuable to the research community.

**Theoretical Claims:**

I have checked the correctness of theoretical claims in the submission.

---

> ### Author Rebuttal · Authors · 2025-04-01
>
> Thank you for recognizing our work as "valuable for improving research efficiency" and providing "valuable insights"! Below, we address the raised weaknesses (W) and questions (Q) with point-to-point clarifications.
>
> > The mentioned new results: https://anonymous.4open.science/r/icml_submission8414-7C89/rebuttal.pdf
>
>
> ---
>
> ## [W1 & Q2] "how the sandbox improves model development compared to other platforms remains unclear." "What are the advantages of the sandbox in model refinement?"
>
> - In the perspective of **development process**: The sandbox is designed for *cost-sensitive* model optimization, enabling broader and more quantitative experiments within the same resource constraints. Compared to naive trial-and-error, this allows *more possibilities* for model improvement with less computational cost.
>
> - In the perspective of **transferable mechanism**: Our sandbox is insight-driven and emphasizes *analysis and scaling laws* derived from extensive experimental feedback. This approach has been validated across 40+ data processing operations and 70+ performance metrics, with empirical success on 5 cross-task, cross-scale models (e.g., EasyAnimate, T2V-Turbo, MGM, CLIP, InternVL).
>
> - **Rapid Expansion & New Experiments**: The framework is highly flexible and extensible:
>   - We added experiments on a new model series, *InternVL (1B, 2B, 4B, 26B)*, to verify generalization in captioning tasks. Details are in our reply to Reviewer CLJN.
>   - We demonstrated iterative training: starting with ckpt0 → refining recipe_1 → training ckpt1 → refining recipe_2 → training ckpt2. Notably, ckpt2 improved performance despite evolving recipes. See our reply to Reviewer 3zp6.
>
> - **Evidence Summarization**:
> Collectively, extensive experiments demonstrate the effectiveness and usability of the sandbox, covering different
>    - *Scenarios*: Text-to-video generation, image-text pre-training, image-to-text generation for general image understanding, and image captioning, iterative enhancement.
>    - *Stages*: Both pre-training and fine-tuning.
>    - *Model Architectures*: EasyAnimate, T2V-Turbo, MGM, InternVL2, and CLIP.
>    - *Scales*: Model parameters, deduplicated data sizes, and compute FLOPs.
>    - *Data Processing Solutions*: Over 40 Data-Juicer Filter OPs and Mappers (see response to Reviewer 5ZDF)
>    - *Model Feedback*: 78 metrics.
>
>
> ---
>
> ## [W2 & Q1] "Data recipes obtained from small-scale experiments may not generalize to large-scale" "I am curious to learn about how the data recipes obtained from this sandbox compare to the optimal data recipes in large-scale experiments."
>
> - **Clarification on Workflow**: Recall that our workflow begins with small-scale experiments (Sec 4.2–4.3) but rigorously validates findings on larger scales (Sec 4.3–4.4). Empirically:
>   - Small-scale recipes benefit larger scales across architectures, data, and compute (see structural summarization in Table 5).
>   - Scaling behavior (Fig.3 & new InternVL exps), high data efficiency (Table 2), and SOTA performance (Table 3, with many *SOTA* large-scale recipes used by baselines) are also validated even under challenging distribution shifts.
>
> - **Methodological Flexibility**: The open-source framework supports custom workflows while reusing data/model infra (Sec B.1–B.3). Advanced strategies like dataset distillation [1] and tensor shaping [2] can be integrated via new hooks. We leave theoretically addressing distribution shifts as future work, which requires substantial new effort.
>
> [1] Large scale dataset distillation with domain shift, ICML'24
>
> [2] Scaling for Training Time and Post-hoc Out-of-distribution Detection Enhancement, ICLR'24
>
> ---
>
> ## [W3 & Appendix] "There is no supplementary material of this submission", "I think the sandbox's properties should be the core focus, but there are many paragraphs talking about observations from data experiment"
>
> We agree that the sandbox's properties are critical and clarify potential misunderstandings:
> - **Extensive Appendix**: Contrary to the comment (you may not have noticed our details), we provide a dedicated appendix (~1.5 pages) focusing on infrastructure:
>   - Overview architecture (Sec B.1),
>   - Capability factory and behavior hooks (Sec B.2),
>   - Extensibility (Sec B.3).
>
> - **Main Paper Focus**: Due to space constraints, we prioritize showcasing practical results and insights in the main text. Without concrete demonstrations, readers may struggle to appreciate the sandbox’s utility and understand its usability. System details are thus deferred to the appendix and source code for a smooth and progressive reading experience.
>
> ---
> Thanks again for your time and valuable comments! We hope that you can consider an increase in the rating if these responses and clarifications address your comments. If you have any additional comments, we warmly invite you to share them with us.

---

> > ### Comment · Reviewer_s5fL · 2025-04-07
> >
> > Thanks for the author's responses. I still have some concerns that the experiments presented in the main paper may not effectively demonstrate the advantages of the proposed sandbox compared to existing approaches-particularly regarding the claimed property of integrated data-model co-development. Nonetheless, I acknowledge the sandbox’s contribution to the multimodal research community and would like to raise my score to 3.

---

> > > ### Author Response · Authors · 2025-04-09
> > >
> > > We appreciate your acknowledgment of the sandbox's contributions! To address the remaining comments on its advantages, we provide clarifications below.
> > >
> > > ---
> > >
> > > ## Key Differentiations from Existing Works
> > >
> > > In Appendix A (lines 672–752), we outlined how the sandbox differs in three areas:
> > >
> > > ### 1. Model-Centric Approaches
> > > Existing efforts focus on refining training algorithms, architectures, and applications. However:
> > > - They rely heavily on scaling laws and optimization techniques, which are computationally expensive and dataset-specific.
> > > - They rarely analyze how data processing impacts downstream performance.
> > >
> > > Our advantage: The sandbox links data processing effects to model performance through systematic experiments, offering actionable insights into data-model interactions.
> > >
> > > ### 2. Data-Centric Approaches
> > > Recent trends emphasize data quality and scale. However:
> > > - These methods isolate data processing from model training.
> > > - They depend on heuristic methods, such as filtering based on human intuition.
> > >
> > > Our advantage: The sandbox provides a systematic framework for data-model co-development, treating both as equally important. For example:
> > > - We analyze correlations between data pools and reference model metrics across tasks.
> > > - We extend beyond CLIP-like models to include LLaVA-like and DiT-based models for broader applicability.
> > >
> > > ### 3. Open-Source Infrastructures
> > > While strong infrastructures exist for model training and evaluation, multimodal data-model co-development remains underdeveloped:
> > > - Current tools focus on single-modal data or dataset-specific preprocessing.
> > > - No dedicated open-source platform exists for foundation model co-development.
> > >
> > > Our advantage: The sandbox integrates cutting-edge model-centric infrastructures with the Data-Juicer system, creating a streamlined environment for co-development.
> > >
> > > ---
> > >
> > > ## Key Differentiations in System Design
> > >
> > > Our design is modular and one-stop. Appendix B details its distinct architecture:
> > > - *Bottom Layer*: Factory classes integrate diverse training/eval libraries.
> > > - *Middle Layer*: Encapsulates model-agnostic behaviors like probing, training, and evaluation hooks.
> > > - *Top Layer*: Implements workflows via ordered job lists, enabling users to adjust configurations via files.
> > >
> > > This design reduces the need to learn disparate libraries or start from scratch, allowing flexible reuse of representative modules:
> > > - Our experiments span 5 architectures, ~80 metrics, and diverse pre-/post-training setups for multimodal models.
> > > - Thanks to its middleware design, the sandbox evolves with underlying libraries, amplifying utility through open-source contributions (e.g., supporting 500+ LLMs and 100+ eval datasets).
> > >
> > > At the code level, we simplify user actions by providing:
> > > - One-line scripts to run and adjust configurations.
> > > - Dynamic dependency resolution and containerization for auto-setup.
> > > - Lazy-loading of Python packages for easy module loading.
> > > - Unified scripts for environment preparation, with third-party modules organized as services.
> > >
> > > ---
> > >
> > > ## Key Differentiations in Extensibility
> > >
> > > Another strength is the sandbox's high extensibility, adaptable to various scenarios. Below are two examples of extending workflows.
> > >
> > > ### Case 1: New Data Requests
> > > - How to Extend: By wrapping a configurable loop (~5–10 lines of code changes), we reuse the "probe-analyze-refine" workflow for iterative model improvements.
> > > - New Example Added During Rebuttal
> > >   - Setting: Starting with *ckpt0*, we refined *recipe1* → trained *ckpt1* → refined *recipe2* → trained *ckpt2*. The first iter aligns with single-OP experiments from previous tasks. We then selected the best checkpoint and applied top OPs to the original dataset as a baseline.
> > >   - Results (summarized in Table 2 of the uploaded PDF):
> > >     - The second round further improves performance.
> > >     - Compared to the first round: 2 new OPs emerge, and OP rankings changed by an average of -3.33~6.14 positions.
> > >     - Using the same OPs, data pool performance changes, providing actionable insights for dynamic environments.
> > >
> > > ### Case 2: Automated Prompt Adjustments
> > > - How to Extend: By adding a model inference hook (~40 lines of code changes) and updating its prompt cfg file, we reuse prior codes to study and optimize prompts.
> > > - New Example Added During Rebuttal:
> > >   - Setting: Using top-1 recipes from two iterations, we studied the effects of 10 different prompts auto-generated by Qwen2.5-max with the prompt: "You are a prompt expert, please optimize the given one: {pmt}."
> > >   - Results (summarized in Table 4 of the uploaded PDF):
> > >     - Prompt rankings and performance vary significantly across iterations (e.g., ranging from -1.673 to +0.944).
> > >     - Some prompts outperform the baseline, while others show inconsistent behaviors.
> > >     - These findings highlight the sandbox's potential for model cfg optimization.
> > >
> > > ---
> > >
> > > Hope the response can clarify its unique pros to advance integrated data-model co-dev. Thank you again for your valuable feedback!

---

### Official Review · Reviewer_3zp6 · 2025-03-23

**Overall Recommendation:** 3

**Summary:**

In their work, authors describe and implement a method and procedures for data-model co-development, aiming on improving pretraining of various foundation model types (language-vision CLIP, diffusion based text-to-video generative models, Llava based image-text generative model). The framework authors introduce is Data Juicer Sandbox. It sets up a pipeline for data composition, making use of available data pools to apply various data processing operators (OP) aiming for data composition that will improve model training. Authors define a probe-analyse-refine workflow, which applies OPs to data pools, employing low cost model training and evaluating those, which provides dataset quality scores. Based on those, OPs leading to top performing evals can be combined to processing pipelines. Those are used to create datasets at larger scales, which should further improve model pre-training. Authors test their approach by performing dataset composition for the aforementioned model types and showing that it leads to strong scores on standardized benchmarks like MMBench, VBench, ImageNet zero-shot classification, such that obtained models compare well to the known strong reference baselines.

**Claims And Evidence:**

Main claim of the study is to be able to perform model-dataset codevelopment. Authors argue that dataset composition performed by their method leads to stronger models by comparing models they obtain to well known references. The comparison is done on fixed compute scales for each model (for video models) or on few reference scales (for CLIP). For full comparison, I think it would require a proper scaling law per each model, as taking fixed reference compute scales might be misleading (eg differences models show for certain fixed scales can change across scales, so trend across scales for each tested model is the actual interesting measure for comparison). Another problematic issue I think is that to claim codevelopment, authors would have to show at least two iteration cycles of dataset composition and model improvement. In current form it seems to me that while the evidence might enough to claim dataset composition via the described procedure leads to training a strong model, it is not clear whether then the obtained model can be used to further improved dataset composition, or at least that the dataset can be further improved from insights of conducted experiments, and so on (which would be the case of co-development).

**Essential References Not Discussed:**

Relevant scientific literature is well covered.

**Experimental Designs Or Analyses:**

The experimental design that describes the dataset composition testing pipeline seems sound to me.

**Methods And Evaluation Criteria:**

Authors make use of a number of standardized benchmarks to measure the model quality, which makes sense.

**Other Comments Or Suggestions:**

A lot of important information on CLIP experiments is in the Appendix. It would be good to have those more visible in the main part, as this is important reference model class and showing improvements there provides strong evidence for the validity of the dataset composition approach.

UPDATE: increasing score to 3 after clarifications on model-data co-development.

**Other Strengths And Weaknesses:**

Strength of the work is variety of different models authors consider and in attempt to define a generic pipeline for dataset composition improvement. Weakness are missing scaling laws that would be necessary to see whether improvements are happening indeed across various scales and not on some fixed reference scale authors happen to choose.

**Questions For Authors:**

Scaling law derivation is performed on smaller scales. Would it be possible to conduct dataset composition at smaller scales and show that compositions made by the approach result in consistent trend improving across scales, without investing much compute?

**Relation To Broader Scientific Literature:**

The work continues tracks set by important previous works like DataComp, emphasizing its difference in attempt to incorporate model signals during dataset composition. It is however I think not correct that previous works were not doing so, eg DataComp was incorporating CLIP models to filter the data, which then can be also considered as a case of model-data codevelopment. As authors do not show many iterations of such a cycle, they do not really push further in approach compared to DataComp, in my opinion (which would be different if authors have demonstrated iterative improvement of model over some cycles of their co-development)

**Theoretical Claims:**

Authors make a theoretical argument to back up the approach for carrying over observations done on small scale pool experiments to larger dataset composition. It seems to me the derivation makes sense.

---

> ### Author Rebuttal · Authors · 2025-04-01
>
> We are grateful for your recognition of the *method, evaluation, theoretical argument & exp design* of our work. Regarding your raised concerns, we address all of them point by point as follows:
>
> > New results added: https://anonymous.4open.science/r/icml_submission8414-7C89/rebuttal.pdf
>
> ---
>
> ### [In Evidence, Weakness & Question parts] "experiments conducted on *per model* with *various scales*; consistent trend improving across scales?"
> Following your suggestion, we conducted additional experiments on a larger model suite (*1B, 2B, 4B, 26B InternVL*) and a new task (*fine-tuning for captioning*). These results further validate the generalizability of our work. For details, please refer to our response to Reviewer cLJN.
>
> ---
>
> ### [In Evidence & Literature parts] "iterative co-development, ... push further than DataComp"
> Thank you for your insightful comments! To address this, we performed new experiments to demonstrate iterative training:
>
> - *Setting*: Starting with `ckpt0`, we refined `recipe_1` → trained `ckpt1` → refined `recipe_2` → trained `ckpt2`. The first iteration aligns with the single-operation (OP) experiments from previous tasks. Based on the first-round results, we selected the best checkpoint and applied the found top OP, *text_length_filter*, to the original training dataset as our base model. We then conducted a second iteration of single-OP continuous training to evaluate its impact.
>
> - *Results*: Summarized in Table 2 of the uploaded PDF, the second iteration continues to improve performance. Notably, while the ranking of OP effects changes slightly between iterations, most effective OPs remain consistent: 8 out of the top-10 OPs in the second iteration overlap with those from the first iteration.
>
> - *Remark*: This highlights promising future directions and underscores the distinction between our work and DataComp. While DataComp primarily focuses on CLIP pretraining and scaling compute with duplicated data, our data recipes are validated across a broader range of tasks, including text-to-video generation (*EasyAnimate, T2V-Turbo*), image-to-text pretraining (*MGM*), post-training (*newly added InternVL in the rebuttal*) and this iterative co-development. For a more detailed comparison in the scaling perspective (including scaling across compute and number of distinct data samples), plz see Fig. 3, Table 5 in the original paper, and the newly added InternVL experiments mentioned above.
>
>
> ---
> ### [Writing Suggestion & Appendix] "have CLIP info more visible in the main part", "one important figure 12 is missing"
> We appreciate these constructive suggestions! Regarding "Figure 12" mentioned in L1424, it should indeed be "Table 12."
> For CLIP-related information, we will move more detailed settings and results analysis from the Appendix into Section 4.1 and the first paragraph of Section 4.5.
>
> ---
>
> **Empirical Evidence Summarization** Collectively, extensive experiments demonstrate the effectiveness and usability of the sandbox, covering different
> - *Scenarios*: Text-to-video generation, image-text pre-training, image-to-text generation for general image understanding, and image captioning, iterative enhancement (see details above).
> - *Stages*: Both pre-training and fine-tuning.
> - *Model Architectures*: EasyAnimate, T2V-Turbo, MGM, InternVL2, and CLIP.
> - *Scales*: Model parameters, deduplicated data sizes, and compute FLOPs.
> - *Data Processing Solutions*: Over 40 Data-Juicer Filter OPs and Mappers (see response to Reviewer 5ZDF)
> - *Model Feedback*: 78 metrics.
>
> ---
>
> Thank you again for your helpful feedback! We will incorporate these improvements into the final version, and kindly request you to review these responses and re-evaluate the merits of this work. Your feedback is highly anticipated and greatly valued.

---

> > ### Comment · Reviewer_3zp6 · 2025-04-03
> >
> > I am pleased to see one of the central  questions on model-data codevelopment character was addressed in insightful manner. I will thus raise the score to 3, as I think the work is now substantial enough to be of interest for ICML community.

---

### Official Review · Reviewer_cLJN · 2025-03-25

**Overall Recommendation:** 4

**Summary:**

This paper introduces Data-Juicer Sandbox, a feedback-driven suite for multimodal data-model co-development. The system integrates the data processing system with model-centric infrastructure, and designs a "Probe-Analyze-Refine" workflow to systematically explore the relationship between data processing operators and model performance. The main contributions of the paper include: (1) the first open-source sandbox platform supporting joint optimization of multimodal data and models; (2) empirical validation demonstrating significant performance improvements in image-to-text generation, text-to-video generation, and image-text pretraining tasks ; (3) scalability verification showing that optimization strategies obtained from small-scale experiments can be transferred to large-scale scenarios, thereby reducing computational costs.

**Claims And Evidence:**

The paper's main claims are supported by experimental results. The authors assert that Data-Juicer Sandbox can optimize data processing strategies through feedback-driven approaches, thereby enhancing model performance, which has been validated through experimental results across multiple tasks.

**Essential References Not Discussed:**

no

**Experimental Designs Or Analyses:**

Experimental results in table 3 showcase the significant performance gain by distillation on T2V-Turbo. There could be additional experiments on other models to explore the broader applicability and benefits of Data-Juicer

**Methods And Evaluation Criteria:**

The proposed methods and evaluation criteria are generally reasonable.

**Other Comments Or Suggestions:**

I suggest adding experiments with models of different scales (from small to large) to better demonstrate the scalability of the method.

**Other Strengths And Weaknesses:**

Strengths:
- The system design has practical value, especially for researchers and engineers who need to rapidly iterate on multimodal model development.
- Experimental results indicate that the method is effective across multiple tasks, demonstrating its versatility.

Weaknesses:
- Baseline comparisons are not comprehensive enough.

**Questions For Authors:**

no

**Relation To Broader Scientific Literature:**

This work is related to the broader research field of multimodal data processing and model training.

**Theoretical Claims:**

The paper does not present theoretical claims requiring rigorous mathematical proof. Its core contribution is a practical system framework and methodology, primarily validated through experimental results.

---

> ### Author Rebuttal · Authors · 2025-04-01
>
> We sincerely appreciate your acknowledgment and constructive feedback on our work! We respond to the raised only concern with the following new experiments.
>
> > The mentioned new results: https://anonymous.4open.science/r/icml_submission8414-7C89/rebuttal.pdf
>
> ---
> ### [Exp Design, Weakness & Suggestion] "additional experiments on other models," "... with models of different scales (from small to large), ..."
> Thank you for your valuable suggestion! Per your comments, we have conducted new experiments on an additional model, *InternVL2.0*, across different scales (1B, 2B, 4B, 26B), covering a new task (*image captioning*) and new stage (*fine-tuning*).
>
>
> `Settings`
>
> To ensure consistency with our proposed workflow:
> - We first experiment with the smallest model size (1B parameters) on *23 single-op* and *6 multi-op data pools*, deriving the optimal data recipe.
> - The identified recipe is then applied to all selected model scales.
>
> Specifically:
> - We fine-tune InternVL2.0 for image captioning using the *COCO Caption training dataset* (567k image-caption pairs).
> - Each single-op data pool contains ~189k samples, while multi-op pools are reduced to ~24k samples due to intersections.
> - Evaluation metrics include *Bleu-1/2/3/4, METEOR, ROUGE_L, and CIDEr*, along with the average performance change relative to baseline models trained on randomly sampled data pools.
>
> For fine-tuning implementations, we set the global batch size to 512. We only take 1 epoch for each data pool with 1B-parameter model. For all experiments, we use H20 GPUs to fine-tune and evaluate the models. Training on one single-op, multi-op data pool with 1B-parameter model takes about 8 and 1.5 GPU hours. Training the optimal recipe including about 24k samples on models with 1B/2B/4B/26B parameters takes about 1.3/2/4/18 GPU hours respectively. Due to time constraints, experiments on various scales using the optimal recipe were repeated twice (random seeds: 42, 1024). Other experiments were conducted once.
>
>
> `Key Results`
>
> 1. *Single-OP Results*:
>    - Full results are ranked in *Table 1* of the updated PDF.
>    - Models trained on shorter captions (*Text Length Low*) achieve the best performance.
>    - Most OPs show positive improvements.
>
> 2. *Multi-OP Results*:
>    - Top-3 OP combinations do not yield further gains, likely due to low correlation between these OPs (plz see *Figure 4* in the updated PDF for Pearson correlation coefficients).
>
> 4. *Scaling Behavior*:
>    - Using the top-3 combination recipes, we fine-tuned InternVL models across all scales.
>    - Results are shown in *Fig.1(b)* of the updated PDF (x-axis in log scale).
>    - *Highlight*: All three recipes maintain consistent and steady performance advantages as the model scale increases from 1B to 26B, demonstrating clear *scaling law behaviors*. In conclusion, our sandbox can consistently achieve performance improvements across these studied scales.
>
> ---
> Thank you again for your time and support! We hope these new results and revisions can address your concern and strengthen the work's applicability. If you have further questions or suggestions, we warmly invite your input.

---

### Decision · Program_Chairs · 2025-05-01

**Decision:**

Accept (spotlight poster)

**Comment:**

This paper presents a framework for multimodal data-model co-development. While reviewers initially raised concerns about scalability and baselines, the authors addressed these through additional experiments. The work demonstrates practical value across multiple architectures and tasks. Reviewers acknowledged the improvements, and the paper's contributions outweigh remaining concerns. ​The AC recommends acceptance of the paper. Congratulations to the authors!